# When Case Gets Rare: A Retrieval Benchmark for Off-Guideline Medical Question Answering

## Abstract

Across medical specialties, clinical practice is anchored in evidence-based guidelines that codify best studied diagnostic and treatment pathways. These pathways work well for the majority of patients but routinely fall short for the long tail of real-world care not covered by guidelines. Most medical large language models (LLMs), however, are trained to encode common, guideline-focused medical knowledge in their parameters. Current evaluations test models primarily on recalling and reasoning with this memorized content, often in multiple-choice settings. Given the fundamental importance of evidence-based reasoning in medicine, it is neither feasible nor reliable to depend on such memorization in practice. To address this gap, we introduce OGCareBench, a long-form retrieval-focused benchmark aimed at evaluating LLMs at answering clinical questions that require going beyond typical guidelines. Extracted from published medical case reports and validated by medical professionals, OGCareBench contains long-form clinical questions requiring free-text answers, providing a systematic framework for assessing open-ended medical reasoning in rare, case-based scenarios. Our experiments reveal that even the best-performing baseline (GPT-o3-mini) correctly answers only 51% of our benchmark with open-source models only reaching 36%. Augmenting the models with retrieved medical articles improves this performance to up to 75% (using GPT-5) highlighting the importance of evidence-grounding for real-world medical reasoning tasks. OGCareBench thus establishes a foundation for benchmarking and advancing both general-purpose and medical language models to produce reliable answers in challenging clinical contexts.

## 1 Introduction

Large language models (LLMs) are actively being explored in healthcare settings for many use cases and hold the potential to transform clinical decision-making and ultimately enhance patient outcomes (Yan et al., 2024; Abrar et al., 2024; Shool et al., 2025). Realizing this potential requires evaluations that reflect the diversity and complexity of real clinical scenarios. Most current benchmarks, however, test models' recall of medical knowledge through exam-style questions (Ben Abacha & Demner-Fushman, 2019; Krithara et al., 2023), typically in multiple-choice settings (Jin et al., 2019; 2021; Pal et al., 2022; Hendrycks et al., 2021; Zuo et al., 2025). While long-form question-answering datasets exist, they are largely patient-oriented and not designed for clinician-facing decision support (Hosseini et al., 2024; Nguyen et al., 2023; Singhal et al., 2023a; Zhu et al., 2020). This leaves an important gap: current evaluations rarely test whether models can generate expert-level, long-form answers that are appropriately grounded in evidence. Evidence grounding is especially crucial in medicine, where clinical guidance evolves rapidly, authoritative references are essential for trust, and patient care often involves rare conditions and atypical presentations. In such settings, memorization alone is insufficient; models must be able to integrate and synthesize knowledge dynamically from external sources to support real-world clinical decision-making.

In this work, we aim to evaluate LLMs in settings that reflect how physicians actually approach complex clinical problems. To do so, a benchmark must satisfy three key properties: (1) it should be grounded in real patient cases reflecting the variability and nuance of actual clinical practice, (2) it should adopt a long-form question-answering (LFQA) format to capture the open-ended reasoning physicians require, as opposed to multiple choice questions, and (3) it should be non-trivial, demanding expert-level domain knowledge, mirroring the complexity of real-world decision-making. Guided by these principles, we focus on simulating scenarios in which physicians must consult external resources to determine appropriate clinical procedures for patients whose cases fall outside standard guidelines or involve rare, off-guideline presentations.

To serve this purpose, we use published medical case reports. Case reports document novel, rare, or unprecedented clinical occurrences such as unusual case presentations, atypical diagnostic mechanisms or non-standard treatments.

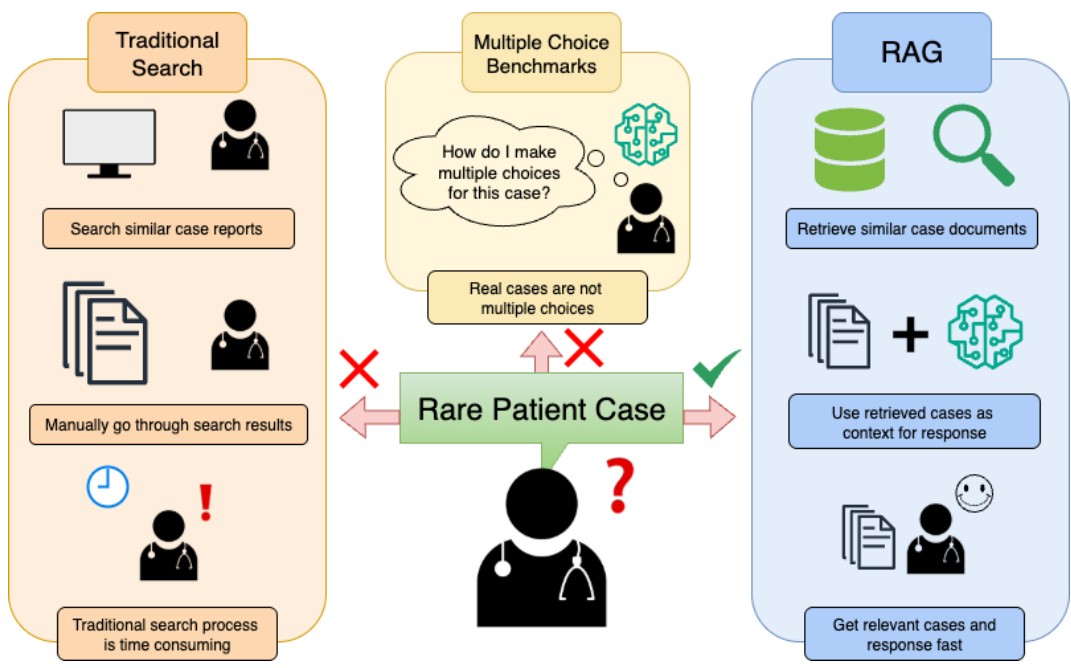

Figure 1: When a physician encounters a rare patient case, using traditional search is time-consuming. Many models currently focus on multiple choice. While there are several models for long-form QA tasks, they have different focuses. We therefore suggest RAG to get clinically reliable responses. OGCAREBENCH serves as a tool for benchmarking RAG in rare patient cases.

Physicians often consult them when typical guideline references, such as UpToDate or standard specialty guidelines (UpToDate, 2025), are insufficient to manage complex or unusual cases. For each case report, we apply semi-automatic methods (§3) to extract a question and answer pair centered around the significant contribution of the report—which could be a novel diagnosis, novel treatment, or a test associated with a rare occurrence of a disease. We refer to this medical benchmark of **O**ff-**G**uideline **Ca**se **Re**ports as OGCAREBENCH. Our dataset contains 235 cases across 10 medical specialties (see Table 1). All questions and answers are validated by experienced physicians to ensure accuracy and fidelity to real-world clinical reasoning.

Our evaluation of several state-of-the-art general-purpose and medical domain-specific models reveals that LLMs struggle to provide expected responses to rare cases. These results highlight the limitations of relying on parametric memory of the models alone when handling rare cases, underscoring the necessity of retrieval augmentation in answering complex medical scenarios. Therefore, we expand our horizon to evaluating performances under retrieval, which is known to enhance the performance of medical question answering (Neha et al., 2025). We create a retrieval corpus of 53,617 case reports covering 12 medical specialties, drawn from publicly available reports. We find providing retrieved documents in the context of the question significantly increases model performance in most models. However, a notable gap exists between the performance of open-source models and proprietary models, and future contributions to enhance open-source models' ability to reason rare cases will be necessary. In summary, we make the following contributions:

- We introduce OGCAREBENCH, a benchmark derived from published medical case reports, designed to evaluate language models on realistic rare clinical scenarios.

- We empirically demonstrate the shortcomings of both medical and general-purpose models in open-ended rare-case reasoning, underscoring the limitations of their standalone use for supporting physicians in real clinical settings.

- We show that retrieval augmentation enhances performance in expert-level tasks, emphasizing its necessity in building robust systems in the medical domain.

## 2 RELATED WORK

**Models and Datasets Focused on Medicine** Medical question-answering (QA) models have significantly evolved (Shool et al., 2025; Yan et al., 2024). A large portion focuses on and is mostly tested on multiple-choice question answering (Han et al., 2025; Wu et al., 2023; Singhal et al., 2023b; Bolton et al., 2024), often using exam-style benchmarks for evaluation (Shool et al., 2025; Krithara et al., 2023; Jin et al., 2019; 2021; Pal et al., 2022; Hendrycks et al., 2021; Zuo et al., 2025). Models and datasets with LFQA style are often patient-oriented (Li et al., 2023; Hosseini et al., 2024; Nguyen et al., 2023; Singhal et al., 2023a; Zhu et al., 2020) or based on general clinical knowledge (Garc'ia-Ferrero et al., 2024; Bolton et al., 2024; Krithara et al., 2023) rather than case-conscious reasoning. Recently, there have been studies focusing on case-based models and dataset (Xu et al., 2025; Nori et al., 2025). Qiu et al. (2025) and Wu et al. (2025) especially focuses on using case reports to construct benchmark on final diagnosis, its reason, and treatments. We broaden the focus and convey the novelty of the case report, whether it may be diagnosis, treatment, or clinical examinations that are presented in a novel way.

**Retrieval augmentation in expert domains** Retrieval augmentation generation (RAG) is known to enhance the performance in knowledge-intensive tasks (Lewis et al., 2021), providing a promising foundation for domain-specific reasoning (Lee et al., 2025). Using RAG in areas requiring domain expertise mitigates the limitation of memorization by integrating curated professional context as shown by examples from legal domain (Zheng et al., 2025; Hou et al., 2024). With medicine, prior studies have shown that incorporating RAG enhances the performance in various medical QA, ranging from multiple choice to case-based reasoning (Xiong et al., 2024; Dong et al., 2025; Ke et al., 2025; Chen et al., 2025). However, use of RAG in various rare-case scenarios and case-based retrieval corpus still remains a gap, and we address this by evaluating rare-case questions using RAG.

## 3 OGCAREBENCH: A BENCHMARK OF OFF GUIDELINE MEDICAL CASES

Medical case reports document novel or rare clinical occurrences. They are typically published to document and highlight unusual conditions, atypical disease courses, unexpected complications, new diagnosis mechanisms, or unique treatment strategies. Case reports appear in specialty journals such as, Journal of Clinical Case Reports, BMJ Case Reports, general medical journals like NEJM, and online repositories. To better understand how case reports are used in practice by physicians, we conducted informal interviews with 10 physicians from different US based institutions with specialties ranging from emergency medicine, rheumatology, internal medicine, infectious diseases, oncology, and surgery. We surmised that while not all practitioners rely on case reports—fields like infectious diseases or emergency medicine rarely need to consult them—specialties such as surgery, internal medicine, and oncology often turn to case reports. Physicians reported that when encountering cases that fall outside standard clinical guidelines[1], they rely on case reports and series alongside consultation with colleagues or specialty networks to identify relevant precedents and guide their clinical decision-making. This is supported by studies showing that only 55% to 57% of guideline-recommended treatments are implemented in routine practice (McGlynn et al., 2003; Runciman et al., 2012).

To construct a dataset that emphasizes such rare, patient-specific cases, we synthesize our benchmark, OG-CAREBENCH, from these reports. Starting from all open-access case reports available on PubMed Central (PubMedCentral, 2003), we filter for cases with novel content and persistent rarity, then extract question-answer pairs using LLMs. To simulate realistic clinical scenarios beyond the scope of the original reports, we apply controlled modifications to these questions, ensuring they are distinct from the documented cases. Finally, all modified questions undergo physician annotation to validate both accuracy and clinical relevance. We outline the benchmark construction in Figure 2 and detail it below. An example of a case report along with the created question-answer pair is provided in Figure 3.

### 3.1 DATASET CREATION

**Step 1: Collecting and filtering case reports** PubMed Central (PMC) provides access to a large collection of open-access medical articles through a File Transfer Protocol (FTP) service, which we use to download relevant articles (PubMedCentral, 2003). To distinguish case reports from other types of medical articles, we leverage the open-access journal list provided by PMC and identify case reports based on the venues in which these journals are published, focusing on those known to regularly feature case report (see Table 15). Using this approach, we compile a total of 53,617 reports, which are then subject to further verification and processing for inclusion in our dataset. We treat this

---

[1]Clinical practice guidelines set by major societies like American College of Cardiology (ACC/AHA), the American College of Rheumatology (ACR), and the National Comprehensive Cancer Network (NCCN), and others codify large bodies of evidence and are regularly updated by broad expert panels.

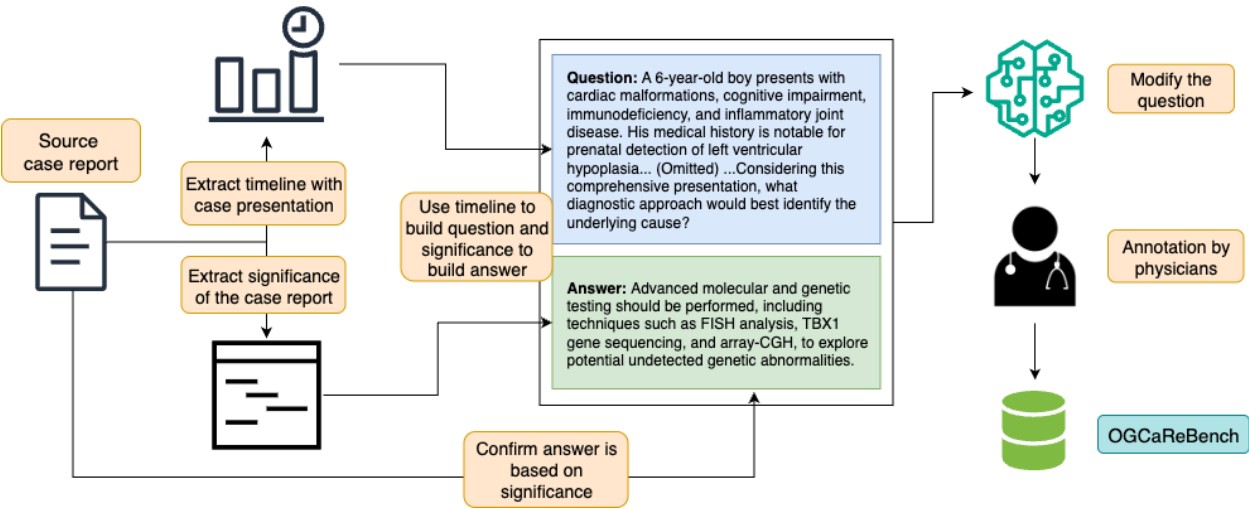

Figure 2: OGCAREBENCH creation pipeline.

corpus as the data store for retrieval-augmented evaluation. To construct a dataset centered on rare cases, we filter this collection to remove reports which meet any of the following criteria: (1) the case report was published in or before 2022. (2) more than three articles cite the case report. (3) the case report is cited by more than one non-case report article. The rationale for these filters is as follows: recently published case reports retain novelty for a few years; case reports cited fewer than a conservative three times are assumed to represent persisting rarity; and case reports cited multiple times by non-case report articles are excluded, as these follow-up studies are indicators that the case has been further explored and potentially resolved into a standard guideline. Based on these criteria, we obtain 28,219 reports.

**Step 2: Extracting raw question-answer pairs** Among the filtered case reports, we randomly select a subset of 1,100 to constitute the dataset. Using GPT-4o (`2024-05-01-preview`), we extract two key elements from each report: (1) the case presentation as a timeline providing a sequence of procedures in patient care, and (2) the most significant contribution of the report, defined as the rationale for its publication. Reports whose significance reflects solely the application of standard procedures to a rare disease, rather than the introduction of a novel intervention, are discarded as the resulting questions and answers will be trivial. The question is then formulated by presenting all procedures preceding the decision point that reflect the significance, asking for the next appropriate step. For example, the most significant contribution of a report might be in developing a new diagnostic test for a condition where all standard diagnostics are inconclusive. In this case, the question will include the patient's history up until the point where the new test was performed and ask what the next step should be. This procedure simulates a scenario where a physician encounters a similar case and has run out of standard guideline-recommended options. The corresponding answer for this question would be the subsequent significant step which in this example is the new diagnostic test. Finally, we use the LLM to verify that the answer was directly connected to the identified significance, thereby confirming the integrity of the question-answer generation process.

Table 1: Distribution of Case Reports across Specialties

| Specialty | All Reports | OGCaReBench |
|---|---|---|
| Basic Sciences | 4437 | 9 |
| Dentistry | 1627 | 3 |
| External Health | 3192 | 12 |
| Intensive Care | 2458 | 9 |
| Internal Medicine | 11530 | 105 |
| Neurology | 1715 | 7 |
| OBGYN | 1601 | – |
| Oncology | 1004 | 7 |
| Orthopedics | 2349 | 17 |
| Pediatrics | 1576 | 6 |
| Surgical Studies | 9408 | 60 |
| General Medicine | 1396 | – |
| Others | 2165 | – |
| Total | 53617 | 235 |

**Step 3: Adding distractors to generated questions** Questions generated in Step 2 are further modified to increase their realism. As our goal is to simulate a situation where physicians consult case reports as guidance for treating their own patients, it is essential that the questions represent unforeseen scenarios and are presented differently from the original reports from which they were derived. To achieve this, we introduce controlled modifications—referred

---

**Stop exsanguination by inflation: management of aorta-esophageal fistula bleeding**
PMCID: PMC10924743

**I. INTRODUCTION**
An aortoesophageal fistula (AEF) is an abnormal connection between the aorta and the esophagus usually secondary to a thoracic aortic aneurysm (TAA) or foreign body ingestion. The presentation typically aligns with Chiari's triad of mid-thoracic pain, sentinel arterial hemorrhage, and exsanguination after a symptom-free interval. Primary AEFs are very rare, with a reported incidence of 0.02% and 0.07% in autopsy studies . . . (Omitted for brevity).

**II. CASE PRESENTATION**
**A 59-year-old male with no known previous medical history presented to a Level 1 trauma center after being discovered by EMS in a large volume of red blood. The event was unwitnessed by bystanders, so it was assumed that the mechanism was a traumatic fall with a resulting head bleed. During transportation patient lost pulses but returned to spontaneous circulation after cardiopulmonary resuscitation. The patient's initial vital signs in the trauma bay were a blood pressure of 129/40 mmHg, a heart rate of 101 beats/min . . .** (Omitted for brevity).

**III. DISUCSSION**
. . . (Omitted for brevity) EMS may assume a traumatic event seeing a massive pool of blood and can further obscure the diagnosis by bandaging the body parts and presenting the story that implies a traumatic mechanism for bleeding. With increasing rates of aortic aneurysms within our aging population, AEFs may preset to trauma personnel. In all successful cases where massive upper gastroesophageal bleeding was stopped, an esophageal balloon or a Sengstaken–Blakemore tube (SBT) was utilized, and this allowed for the temporary stabilization and had bought time for exact CT diagnosis and surgical intervention . . . (Omitted for brevity).

**QUESTION**
A 61-year-old gentleman with no significant past medical history was discovered by paramedics surrounded by hemorrhage, with an unobserved incident thought to result from an accidental tumble causing intracranial hemorrhage. During transport, the patient became pulseless . . . (Omitted for brevity) . . . Management included somatostatin analog administration and activation of massive hemorrhage protocol with 2 units whole blood and 15 packed RBCs transfused. The patient also received empiric antibiotics given concern for aspiration. Cardiac surgery and IR were urgently consulted. What is the most appropriate immediate intervention for hemorrhage control?

**ANSWER**
Inflation of an esophageal balloon or Sengstaken–Blakemore tube to tamponade the bleeding.

Figure 3: Example of case report and corresponding final question-answer pair. Timeline is **bolded** and significance is in blue. Abstract before the introduction and conclusion after the discussion are omitted for brevity and irrelevance. The question asks the direct next step given the patient details (marked red), and the answer is related to the significant point of the case report. Link for full text: `https://pmc.ncbi.nlm.nih.gov/articles/PMC10641966/`.

to as distractions—using Claude 4 Opus. These modifications include altering patient demographics (e.g., age and ethnicity), substituting medical terminology with semantically equivalent expressions, integrating comorbidities that do not affect the original condition, and other related adjustments (see Figure 9 for the full prompt). Importantly, while the questions are modified, the answers are preserved; distractions are applied only to the extent that the clinical plausibility of the case remains intact and the correct answer remains unaffected. To validate this process, three physicians specializing in internal medicine are presented with subsets of the original question, modified question, and corresponding answer. Their evaluation confirms the medical coherence of the modifications. Such distrations mirror the challenges physicians face in real-world settings where unrelated comorbidities may exist.

**Step 4: Dataset verification by experts** We assess the medical validity of the question-answer pairs through annotations provided by three physicians from Step 3. The experts are presented with the modified questions and asked to evaluate them. The evaluation criteria are as follows: (1) the question and answer should be medically aligned, and (2) the question should require domain-specific medical expertise rather than general medical knowledge held by the public. We ask them to rate the pairs on a scale of 1 to 5—1 indicating that question-answer pair is not realistic under any circumstances and 5 indicating that the question is realistic and the answer correctly addresses the question. Only question-answer pairs rated 4 or 5 are retained, yielding 235 instances in the final dataset. Detailed instructions we provide the physicians are in Figure 12.

## 3.2 DATA STATISTICS

We summarize the dataset statistics split across medical specialties in Table 1. We use the original corpus for all 53617 case reports extracted in Step 1 as our retrieval store. The case reports collected are represented by 12 disciplines: basic sciences, dentistry, external health, intensive care, internal medicine, neurology, OBGYN, oncology, orthopedics, pediatrics, surgical studies, and general medicine. Both the full set and OGCAREBENCH are heavily inclined toward Internal Medicine and Surgical Studies. For internal medicine, this is due to its overlap with other specialties and it also encompassing a variety of sub-disciplines such as hepato-biliary-pancreatic and vascular medicine. For surgical studies, each case is unique and hence more case reports are written about them. While there is no instance of OBGYN in OGCAREBENCH, there are surgical studies cases involving maternity care in the dataset. Each question in OGCAREBENCH consists of 1-2 paragraphs, and answers are often 1-2 sentences (length distribution is reported in Table 2).

## 3.3 EVALUATION METRIC

To evaluate the performance of a model using OGCAREBENCH, we feed the question to the model with an instruction to generate a free form natural language answer. To evaluate the alignment between the gold answers extracted from the case reports and the responses generated by the models, we use an LLM-as-a-judge to assess equivalence (specifically, we use GPT-4o). In our early experiments, we find that model responses have varying formats and lengths, ranging from brief phrases to long paragraphs that include background and rationale. To focus on the main clinical content, we prompt the judge with a few-shot example to output "equivalent" or "mismatch" (see Figure 11 for full prompt). Model response is judged as equivalent if the primary clinical procedure recommended matches the procedure specified in the gold answer. Conversely, a response is considered a mismatch if the contents do not overlap or the gold answer appears in the output but not prioritized as the main procedure. Similarly, broad or vague recommendations are labeled as mismatches when the gold answer requires a specific clinical procedure, as our benchmark emphasizes detailed, case-based clinical reasoning. Our primary metric is a simple percentage of answers in the benchmark predicted correctly by the model.

Table 2: Token statistics for the retrieval corpus and queries and answers from OGCAREBENCH (Computed using Contriever's tokenizer (Izacard et al., 2022)).

| Category | Avg. tokens | Max tokens |
|---|---|---|
| Corpus | 2730.21 | 24271 |
| Question | 403.26 | 696 |
| Answer | 29.48 | 120 |

## 4 EVALUATION SETUP

We consider two evaluation setups, (1) a baseline setup in which an LLM is expected to rely on its own parametric knowledge without any retrieval, and (2) a setup where we first perform retrieval on our datastore to find the most relevant case reports and provide the retrieved documents to the model's context to generate the answer. For the retrieval augmented generation (RAG) setup, we first validate the performance of different retrieval models and use the top-performing ones for final evaluation.

Table 3: Maximum context lengths of a subset of models we evaluate with RAG.

| Model | Context Length |
|---|---|
| GPT-5 | 400K |
| GPT-o3-mini | 200K |
| Llama 3.3 70B Instruct | 128K |
| Claude 4 Sonnet | 1M |
| Thinking Claude 4 Sonnet | 1M |
| MedGemma-27b-text-it | 128K |
| Llama 3-Med42-70B | 8K |
| OpenBioLLM-Llama 3-70B | 8K |

## 4.1 BASELINE EVALUATION

We benchmark both state of the art general-purpose and medical domain models as baselines. For general-purpose models, we evaluate: GPT-o3-mini (OpenAI, 2025c), GPT-5 (OpenAI, 2025b), Llama 3.3 (70B Instruct) (Meta AI, 2024), Claude 4 Sonnet (Anthropic, 2025), and Thinking Claude 4 Sonnet (Anthropic, 2025). For models specializing in medical question answering, we evaluate: OpenbioLLM-Llama 3 (70B) (Ankit Pal, 2024), MedGemma (27B-text) (Sellergren et al., 2025), and Llama 3-Med42 (70B) (Christophe et al., 2024). In addition, we also evaluate GPT-4o-search-preview (OpenAI, 2025a) which, as the name suggests, is search-enabled in that it relies on web search (instead of our retrieval datastore) to generate the final answers. We include this baseline to attest to the importance of our retrieval store in solving this benchmark. We prompt the models to answer the question with one best answer (see Figure 10a). Restrictions such as not outputting thoughts and a word limit are added to medical QA models to avoid having unusually lengthy output (see Figure 10b). We also considered Meditron (70B) (Chen et al., 2023) and Clinical Camel (70B) (Toma et al., 2023), but we dropped them since their primary focus is multiple-choice question answering and therefore generated responses by always

outputting answer choices when given questions from OGCAREBENCH (even though no choices were provided in the question).

## 4.2 RETRIEVAL AUGMENTED EVALUATION

**Evaluating Retrieval Methods**   To identify the most effective retrieval models for our downstream generation task, we evaluate a comprehensive set of 15 models encompassing sparse, general purpose, and biomedical models. For the sparse baseline, we employ BM25 (Robertson & Zaragoza, 2009), a model known for its strong performance across various benchmarks, including BEIR (Thakur et al., 2021). Our general-purpose models include All-MiniLM-L12-v2[2], E5-small-v2 (Wang et al., 2024), Contriever and Contriever-MSMARCO (Izacard et al., 2022), and the BGE family (Xiao et al., 2024), which integrates dense, sparse, and multi-vector strategies. For the biomedical domain, we assess MedCPT (Jin et al., 2023), PubMedBERT (Gu et al., 2021), MedEmbed series (Balachandran, 2024), and BMRetriever(Xu et al., 2024), a medically pre-trained and fine-tuned instruction-following model. We also experiment with two-stage retrieval process. Following the initial retrieval, we rerank the top 100 candidates using the PubMed-pretrained MedCPT-cross-encoder (Jin et al., 2023), which has demonstrated state-of-the-art performance on biomedical information retrieval tasks. To assess performance of retrieval, we report results using Recall@k, MRR, and nDCG with respect to the ground-truth case report (from which the question and answer are derived), which together capture different aspects of retrieval effectiveness. Instruction used for BMRetriever is in Figure 5.

Given the long lengths of our corpus and queries, along with the context-window limitations summarized in Table 2 and Table 3, we employ a text processing strategy to optimize document representation. Documents are chunked with a maximum length of 512 tokens and a stride of 128. We then aggregate passage-level scores using a two-level Maximum Passage (MaxP) strategy (Dai & Callan, 2019). The effectiveness of this chunking approach is further validated in Appendix A.2.

**Retrieval Augmented Generation**   We select the best-performing retrievers (see Table 5) from each of the three categories—sparse, general purpose, and biomedical-BM25, BGE, and BMRetriever—to incorporate into the retrieval augmented evaluation. Each of the seven LLMs used in baseline experiments is integrated into the pipeline, except for GPT-4o-search-preview, which incorporates web-search by default. We evaluate the model performance using the top 1, 3, and 5 retrieved case reports as context, as well as an oracle setting in which the ground-truth source case report of the question is input. For OpenbioLLM and Llama 3-Med42-70B, which have a small context window of 8K tokens, case reports exceeding the limit are truncated from the end. Given that the average length of our case reports is 2,730 tokens (see Table 2), including five reports as context for these two models would make a similar setting as using three after truncation. Consequently, we do not test the 5-retrieved reports for them.

## 5 RESULTS AND FINDINGS

**Baselines without retrieval struggle**   Table 4 shows the baseline performance of the base models evaluated with OGCAREBENCH without retrieval augmentation. To validate our LLM-based evaluation, we randomly select 45 baseline results evenly spread across GPT-o3-mini (best performance), MedGemma (lowest performance), and Llama3-Med42 (mid level performance) to be validated by internal medicine physicians. We task them to label whether the GPT evaluation of matching model-generated answers and gold answers reflects true clinical judgment, yielding an agreement of 93%.

We find that general-purpose models overall outperform medical specialized ones. A reasoning model GPT-o3-mini performs the best (51.5%), followed by GPT-5 and Llama 3.3. Claude 4 Sonnet and its reasoning variant trail behind. Surprisingly, GPT-4o-search-preview performs the lowest among general-purpose models at 39.1%, entailing that its built-in search often fails to find the right document to refer

Table 4: Overall baseline performance. Subfield-level results are provided in Appendix A.1

| Model | Accuracy |
|---|---|
| GPT-5 | 44.7 |
| GPT-o3-mini | 51.5 |
| GPT-4o-search-preview | 39.1 |
| Llama 3.3 70B Instruct | 45.1 |
| Claude 4 Sonnet | 41.7 |
| Thinking Claude 4 Sonnet | 40.9 |
| MedGemma-27b-text-it | 36.2 |
| Llama 3-Med42-70B | 42.1 |
| OpenBioLLM-Llama 3-70B | 39.6 |

to. Despite domain specialization, medical LLMs perform poorly with MedGemma, the latest offering from Google, on the lower end among all models at 36.2%. These results show that both state-of-the-art general-purpose models and models for medical tasks struggle when presented with complex rare medical questions. Subpar baseline performance suggests that memorization from pretraining alone is insufficient for handling such cases. Performance of GPT-5 and

---

[2]`https://huggingface.co/sentence-transformers/all-MiniLM-L12-v2.`

GPT-o3-mini suggests that reasoning could offer some advantage in handling rare, case-based scenarios, while Thinking Claude 4 Sonnet doesn't follow this trend. We also speculate that OpenAI's models' performance could be due to recent efforts in improving health-related information communication (OpenAI, 2025b) which might include training with domain specific data. Open-source LLMs in particular remain further behind compared to the proprietary models.

Table 5: Retrieval results across state of the art retrievers. Values are in percentage.

| Type | Model | Params | Recall@1 | Recall@3 | Recall@5 | Recall@10 | Recall@100 | Recall@1000 | MRR@5 | nDCG@10 |
|---|---|---|---|---|---|---|---|---|---|---|
| Sparse | BM25 | N/A | 49.4 | 60.0 | 65.1 | 72.8 | 88.5 | 97.0 | 55.5 | 60.4 |
| General Purpose | All-MiniLM-L12-v2 | 33.4M | 19.1 | 29.4 | 31.9 | 39.1 | 68.1 | 92.3 | 24.3 | 28.5 |
| | E5-small-v2 | 33.4M | 33.2 | 42.6 | 47.2 | 53.6 | 78.7 | 94.0 | 38.6 | 42.8 |
| | Contriever-msmarco | 110M | 34.9 | 46.4 | 51.1 | 59.6 | 80.4 | 94.0 | 41.1 | 46.3 |
| | Contriever | 110M | 34.9 | 46.8 | 53.2 | 60.0 | 78.7 | 91.5 | 41.6 | 46.6 |
| | BGE-small-en-v1.5 | 33.4M | 43.4 | 56.2 | 63.0 | 71.9 | 92.3 | 98.3 | 50.2 | 56.3 |
| | BGE-base-en-v1.5 | 109M | 49.4 | 61.7 | 68.1 | 77.4 | 91.1 | 98.7 | 56.3 | 62.3 |
| | **BGE-large-en-v1.5** | 335M | **61.7** | **74.8** | **80.9** | **86.0** | **95.3** | 98.3 | **69.2** | **73.8** |
| | BGE-m3 | 560M | 44.3 | 56.2 | 58.3 | 66.0 | 89.4 | 97.4 | 50.1 | 54.6 |
| Finetuned | MedCPT | 109M | 15.7 | 26.0 | 31.1 | 37.0 | 66.0 | 92.8 | 21.2 | 25.7 |
| | Pubmedbert-base-embeddings | 109M | 33.6 | 46.3 | 52.3 | 61.3 | 88.5 | 97.0 | 40.7 | 46.5 |
| | MedEmbed-small-v0.1 | 33.4M | 37.0 | 48.5 | 56.2 | 68.1 | 88.5 | 98.7 | 43.7 | 50.5 |
| | MedEmbed-base-v0.1 | 109M | 42.6 | 58.7 | 63.8 | 72.3 | 90.3 | **99.1** | 50.9 | 54.1 |
| | MedEmbed-large-v0.1 | 335M | 51.9 | 64.6 | 68.9 | 75.7 | 92.3 | 97.9 | 58.8 | 63.5 |
| | BMRetriever-410M | 410M | 56.6 | 70.2 | 73.2 | 78.7 | 93.6 | **99.1** | 63.3 | 67.6 |

**Retrieving the right document is hard for complex medical queries** Table 5 presents the retrieval performance of state-of-the-art retrievers. The results demonstrate that OGCAREBENCH is a challenging retrieval benchmark, with only two models achieving a Recall@1 above 50%. For RAG systems, this indicates that the retrieved context is likely to miss crucial information more than half of the time, thereby reducing the quality and accuracy of generated answers. Although Recall@k approaches 100% at very high values of $k$ (100–1,000), providing such a large number of documents as context to LLMs is impractical. Results with simple truncation are reported in Appendix A.5.

As shown in Table 6, incorporating a strong biomedical reranker, MedCPT-Cross-Encoder, consistently improves the performance of most retrieval models. However, reranker leads to performance degradation for several larger retriever models, suggesting that simply adding a specialized reranker is not a guaranteed solution. In some cases, it may even reduce the quality of the final retrieved context, potentially resulting in poorer LLM responses. Results for a general-purpose reranker are provided in Appendix A.6.

**Retrieval augmentation improves results for large context models, but gap remains in others.** Retrieval augmentation generation (RAG) produces substantial performance gains across nearly all evaluated LLMs. GPT-5 and Thinking Claude 4 Sonnet are two models with the most remarkable performance across all three context sizes. In particular, GPT-5 combined with top five retrieved

Table 6: Percentage improvement in retrieval using a reranker (MedCPT-Cross-Encoder).

| Model | Recall@10↑ | nDCG@10↑ |
|---|---|---|
| All-MiniLM-L12-v2 | 66.5 | 88.4 |
| E5-small-v2 | 28.5 | 35.7 |
| Contriever-msmarco | 17.8 | 20.1 |
| Contriever | 15.7 | 20.2 |
| BGE-small-en-v1.5 | 14.7 | 15.6 |
| BGE-base-en-v1.5 | 6.0 | 8.9 |
| BGE-large-en-v1.5 | 1.2 | 2.1 |
| BGE-m3 | -3.0 | -11.0 |
| MedCPT | 39.2 | 63.0 |
| Pubmedbert-base-embeddings | 19.4 | 28.8 |
| MedEmbed-small-v0.1 | 9.4 | 18.2 |
| MedEmbed-base-v0.1 | 3.0 | 10.9 |
| MedEmbed-large-v0.1 | 0.7 | -0.5 |
| BMRetriever-410M | -1.7 | -6.8 |

case reports as context with BGE-Large as the retrieval model achieves the highest accuracy at 75.3%, surpassing the oracle performance of the weakest model (OpenBioLLM). General-purpose reasoning models (GPT-5, GPT-o3-mini, Thinking Claude 4 Sonnet) consistently outperform other models. Among medical models, MedGemma exhibits competitive performance and notable improvement, and Llama 3-Med42 performs comparably. OpenBioLLM exhibits notably subpar performance, especially when using three case reports as context. With this result, we find three important aspects that affect the performance of the model in rare-case scenarios:

- **Retrieval performance is a critical factor influencing RAG performance.** In most settings, BGE-Large backed LLMs deliver the best performance among the three retrievers tested. When comparing the best performing language model across retrievers, using BGE-Large yields the highest accuracy, reflecting its highest retrieval quality.

Table 7: Performance of RAG with different retrieval methods and context length. $\Delta$ stands for the percentage improvement from baseline. For retrieval with BGE, BGE-large-en-v1.5 was used.

| # Reports | Model | BM25 | $\Delta$ BM25 | BGE | $\Delta$ BGE | BMRet. | $\Delta$ BMRet. | Oracle |
|---|---|---|---|---|---|---|---|---|
| | GPT-5 | **63.4** | **18.7** | 67.2 | 22.5 | **68.5** | **23.8** | **88.1** |
| | GPT-o3-mini | 61.7 | 10.2 | 67.2 | 15.7 | 63.0 | 11.5 | 81.3 |
| | Llama 3.3 70B Instruct | 58.7 | 13.6 | 64.7 | 19.6 | 57.0 | 11.9 | 79.6 |
| | Claude 4 Sonnet | 57.9 | 16.2 | 65.1 | 23.4 | 63.8 | 22.1 | 81.7 |
| 1 | Thinking Claude 4 Sonnet | 59.6 | **18.7** | **68.9** | **28.0** | 63.4 | 22.5 | 82.6 |
| | MedGemma-27b-text-it | 51.5 | 15.3 | 62.6 | 26.4 | 56.2 | 20.0 | 78.7 |
| | Llama 3-Med42-70B | 54.9 | 12.8 | 61.3 | 19.1 | 55.7 | 13.6 | 80.4 |
| | OpenBioLLM-Llama 3-70B | 49.4 | 9.8 | 57.9 | 18.3 | 51.5 | 11.9 | 72.8 |
| | GPT-5 | **68.9** | 24.2 | 72.8 | 28.1 | **72.3** | **27.6** | – |
| | GPT-o3-mini | 65.5 | 14.0 | 71.9 | 20.4 | 68.9 | 17.4 | – |
| | Llama 3.3 70B Instruct | 54.5 | 9.4 | 66.0 | 20.9 | 61.7 | 16.6 | – |
| 3 | Claude 4 Sonnet | 56.6 | 14.9 | 67.7 | 26.0 | 64.3 | 22.6 | – |
| | Thinking Claude 4 Sonnet | 66.0 | **25.1** | **73.6** | **32.7** | 66.4 | 25.5 | - |
| | MedGemma-27b-text-it | 51.5 | 15.3 | 63.0 | 26.8 | 55.3 | 19.1 | – |
| | Llama 3-Med42-70B | 51.9 | 9.8 | 56.2 | 14.0 | 53.2 | 11.1 | – |
| | OpenBioLLM-Llama 3-70B | 40.4 | 0.8 | 48.1 | 8.5 | 37.9 | -1.7 | – |
| | GPT-5 | **70.6** | **25.9** | **75.3** | 30.6 | **70.2** | 25.5 | – |
| | GPT-o3-mini | 66.8 | 15.3 | 71.5 | 20.0 | 69.8 | 18.3 | – |
| 5 | Llama 3.3 70B Instruct | 57.4 | 12.3 | 63.8 | 18.7 | 60.0 | 14.9 | – |
| | Claude 4 Sonnet | 62.1 | 20.4 | 67.7 | 26.0 | 66.0 | 24.3 | – |
| | Thinking Claude 4 Sonnet | 66.0 | 25.1 | 73.6 | **32.7** | 67.7 | **26.8** | - |
| | MedGemma-27b-text-it | 43.8 | 7.6 | 56.2 | 20.0 | 50.6 | 14.5 | – |

BMRetriever generally follows, achieving higher accuracy overall compared to BM25, aligning with their relative retrieval performance.

- **Context window size, as well as the number of documents incontext affects the performance.** Models with limited context capacity (Llama 3-Med42, OpenBioLLM) of 8K tokens exhibit the lowest gains, particularly when using three case reports as context. In contrast, MedGemma shows significant improvements with RAG due to its large context window, even though it has the lowest baseline performance. Moreover, different models reach their own peak at different context lengths, suggesting the tradeoff between a long context to process and ensuring the answer's presence among the retrieved documents.

- Finally, **model's reasoning ability effects RAG performance.** GPT-5 and Thinking Claude 4 Sonnet achieve the best result across all retrievers and context lengths, underscoring the importance of reasoning in consulting external sources. Although Thinking Claude 4 Sonnet does not have a notable baseline performance, it demonstrates remarkable improvement when augmented with retrieval, achieving 32.7% gain with five case reports as context and BGE-Large as retriever. GPT-o3-mini also performs competitively, while its improvement is small due to its already high baseline. These patterns highlight crucial reasoning capacity for transforming retrieved content into clinically sound responses.

Overall, RAG improves the performance of the models and thus proves essential for rare-case reasoning, transforming subpar baseline performance into clinically significant results. Full RAG result is provided in Table 7.

## 6 CONCLUSION

Our work suggests that reliable medical LLMs must move beyond memorization and towards benchmarks that reflect real-world clinical reasoning. OGCAREBENCH highlights rare, case-based scenarios where current models fall short. RAG fills this gap by curating the cases to focus on, exhibited by significantly enhanced performance. Together, OGCAREBENCH shows retrieval as a crucial component for building clinically reliable LLMs and establishes a new benchmark for supporting physicians when faced with uncommon clinical cases. Retrieval performance, context window, number of documents used as context, and the reasoning ability all play essential roles when it comes to RAG. We hope this benchmark expands the field of open-ended rare-case reasoning in the medical domain and thereby supports physicians.

## REPRODUCIBILITY STATEMENT

We provide the OGCAREBENCH dataset as supplementary material in csv and json format. "Title" is the title of the source case report that the question-answer pair was derived from, "pmc_id" is PMC ID of the source case report, and "Classification" indicates its specialty. The prompts for dataset construction process in §3.1 are in Appendix C, including significance and timeline extraction, question-answer pair creation, controlled modification, model prompts for evaluation, and evaluating answer and model response equivalence. The full dataset and code will be publicly released upon publication.

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

APPENDIX

LIMITATIONS

OGCAREBENCH has a few limitations. First, the case reports are published and updated regularly, and it is likely that our filtered subset of 28,219 reports, as well as the cases included in OGCAREBENCH, will no longer satisfy the filter criteria as medical knowledge advances. This might make the rare cases part of the guideline reducing the utility of the benchmark in a few years. Therefore, future efforts may require updating the datasets or developing a dynamic version of OGCAREBENCH to mitigate data depreciation. This issue of benchmark saturation is however not unique to medical domain and has been explored extensively in the literature. It has additional challenges in medicine as expert annotation is required. Second, case reports vary in quality; some recommend the use of a specific product on behalf of a company. While we exclude such instances in OGCAREBENCH, they remain in the retrieval corpus. This may lead to the recommendation of certain products during RAG. Third, certain questions may have more than one clinically appropriate next step, and it is feasible that those are not captured by our answer. Although physician validation reduces this risk, multiple answers may still exist in some instances, given the complex nature of clinical practice. Finally, there are other venues to handle rare cases, such as platforms only accessible by physicians. They are the edge cases that are not covered by using case reports as our sources, and in practice RAG will not be able to solve them, as it is not included in any publicly available retrieval corpus.

DISCLAIMER

OpenBioLLM is included as one of our baselines as it has been included in multiple prior studies (Shoham & Rappoport, 2024; Dorfner et al., 2024). However, the model is released without an accompanying paper, data description, or detailed methodological description. Therefore, its performance should be interpreted with consideration and we include it for completeness and comparability with existing benchmarks.

## A ABLATION STUDY

### A.1 BASELINE IN SUBFIELDS

Table 8: Baseline performance across surgery studies and internal medicine.

| Model | Surgery | Internal |
|---|---|---|
| GPT-5 | 40 | 48.6 |
| GPT-o3-mini | 50 | 54.3 |
| GPT-4o-search-preview | 43.3 | 40 |
| Llama 3.3 70B Instruct | 35 | 46.7 |
| Claude 4 Sonnet | 41.6 | 42.9 |
| Thinking Claude 4 Sonnet | 43.3 | 43.8 |
| MedGemma-27b-text-it | 36.6 | 40 |
| Llama 3-Med42-70B | 43.3 | 45.7 |
| OpenBioLLM-Llama 3-70B | 40 | 38.1 |

Table 8 shows baseline results for two major disciplines: surgical studies and internal medicine. Most models exhibit better accuracy for internal medicine, with the exception of GPT-4o-search-preview and OpenBioLLM. GPT-5 and Llama 3.3 have a significant performance gap of over 8%.

### A.2 COMPARISON OF CHUNKING METHODS

Table 9 presents the results of our chunking experiments on two top-performing retrievers, BGE-large and BMRetriever. We compared different combinations of chunking and truncation. Our chunking strategy used a maximum length of 512 tokens with a stride of 128, while truncation was a simple cut-off at 512 tokens. The results demonstrate that applying chunking to both the corpus and the query is essential for achieving high performance in our use case.

Table 9: Retrieval results using different chunking methods.

| Model | Corpus | Query | Recall@1 | Recall@3 | Recall@5 | Recall@10 | Recall@100 | Recall@1000 | MRR@5 | nDCG@10 |
|---|---|---|---|---|---|---|---|---|---|---|
| BGE-large-en-v1.5 | chunk | chunk | **61.7** | 74.8 | **80.9** | **86.0** | **95.3** | 98.3 | **69.2** | **73.8** |
| | chunk | truncation | 60.9 | **74.9** | **80.9** | **86.0** | **95.3** | 98.3 | 68.7 | 73.4 |
| | truncation | truncation | 35.7 | 52.8 | 60.9 | 67.2 | 88.5 | **98.7** | 45.4 | 49.2 |
| BMRetriever-410M | chunk | chunk | **56.6** | **70.2** | **73.2** | **78.7** | **93.6** | **99.1** | **63.3** | **67.6** |
| | chunk | truncation | 55.3 | 70.1 | 72.8 | **78.7** | 93.2 | 98.7 | 62.1 | 66.7 |
| | truncation | truncation | 31.9 | 46.4 | 53.6 | 61.3 | 80.9 | 96.6 | 39.9 | 45.8 |

Table 10: Retreival results using different context lengths.

| Model | Max Len | Recall@1 | Recall@3 | Recall@5 | Recall@10 | Recall@100 | Recall@1000 | MRR@5 | nDCG@10 |
|---|---|---|---|---|---|---|---|---|---|
| BMRetriever-410M | 512 | **56.6** | **70.2** | **73.2** | **78.7** | **93.6** | **99.1** | **63.3** | **67.6** |
| | 1024 | 50.6 | 63.0 | 68.1 | 74.5 | 87.7 | 97.9 | 57.3 | 62.1 |
| | 2048 | 26.4 | 45.1 | 50.2 | 57.9 | 82.6 | 93.2 | 36.2 | 42.3 |

## A.3 EFFECTS OF CONTEXT LENGTH

To evaluate the impact of context length, we conducted an experiment with BMRetriever, which supports a maximum context length of 2,048 tokens and was tested with a fixed stride of 128. The results, presented in Table 10, indicate that merely increasing the context window does not necessarily yield improved performance—particularly for long, domain-specific medical queries such as those in our dataset.

## A.4 EFFECT OF STRIDE VALUES

Table 11: Retrieval results using different stride values.

| Model | Stride | Recall@1 | Recall@3 | Recall@5 | Recall@10 | Recall@100 | Recall@1000 | MRR@5 | nDCG@10 |
|---|---|---|---|---|---|---|---|---|---|
| BGE-large-en-v1.5 | 128 | **61.7** | **74.9** | **80.9** | **86.0** | **95.3** | 98.3 | **69.2** | **73.8** |
| | 256 | 54.0 | 68.5 | 76.6 | 81.3 | 94.5 | **98.7** | 62.6 | 67.6 |
| | 384 | 51.1 | 65.5 | 70.2 | 76.2 | 94.9 | **98.7** | 58.7 | 59.5 |
| | 512 | 47.7 | 63.0 | 69.4 | 77.9 | 91.5 | 98.3 | 56.1 | 62.1 |
| BMRetriever-410M | 128 | **54.5** | **68.5** | **72.8** | **78.7** | 93.2 | 98.7 | **61.8** | **66.4** |
| | 256 | 53.2 | 64.3 | 71.1 | 76.6 | **93.6** | **99.1** | 59.8 | 64.3 |
| | 384 | 48.1 | 61.3 | 70.2 | 74.9 | 90.2 | 98.7 | 56.1 | 61.1 |
| | 512 | 43.4 | 57.0 | 63.0 | 67.7 | 88.9 | 97.4 | 50.7 | 55.2 |

To see the effect of different stride values, we conducted experiments on the two top-performing retrievers, BGE-large and BMRetriever models, with a fixed maximum context length of 512. The results, as detailed in Table 11, revealed that a stride of 128 consistently outperformed other configurations. Consequently, this stride value was selected for all subsequent experiments.

## A.5 RETRIEVAL RESULT UNDER SIMPLE TRUNCATION

Table 12 reports retrieval performance under a simple truncation strategy with a maximum context length of 512 tokens for both corpus and queries. As expected, performance is consistently lower than with the chunking strategy, with no model achieving Recall@1 above 50%. This highlights underscore the importance of chunking and reveal substantial room for improvement in modern retrievers, particularly for rare-case retrieval.

## A.6 RETRIEVAL RESULTS USING GENERAL PURPOSE RERANKER.

Table 13 reports the performance of the general-purpose BGE-reranker-large (Xiao et al., 2024) when applied to the top-100 candidates. Consistent with the findings presented in the main paper, BGE-reranker-large exhibits a notable decline in performance. This result highlights even state-of-the-art rerankers struggle to perform effective reranking within the context of OGCAREBENCH.

Table 12: Retrieval results using simple truncation.

| Type | Model | Params | Recall@1 | Recall@3 | Recall@5 | Recall@10 | Recall@100 | Recall@1000 | MRR@5 | nDCG@10 |
|---|---|---|---|---|---|---|---|---|---|---|
| Sparse | BM25 | N/A | **49.4** | **60.0** | **65.1** | **72.8** | **88.5** | 97.0 | **55.5** | **60.4** |
| General Purpose | All-MiniLM-L12-v2 | 33.4M | 5.1 | 9.4 | 11.5 | 15.7 | 41.3 | 68.1 | 7.4 | 9.8 |
| | E5-small-v2 | 33.4M | 13.2 | 23.4 | 28.5 | 32.8 | 59.6 | 83.0 | 18.9 | 22.9 |
| | Contriever-msmarco | 110M | 13.2 | 25.5 | 29.8 | 35.7 | 64.7 | 85.5 | 19.3 | 23.8 |
| | Contriever | 110M | 15.7 | 24.7 | 27.2 | 32.8 | 51.5 | 72.3 | 20.2 | 23.8 |
| | BGE-small-en-v1.5 | 33.4M | 31.9 | 46.0 | 51.5 | 61.3 | 83.0 | 95.3 | 39.2 | 45.3 |
| | BGE-base-en-v1.5 | 109M | 22.6 | 37.9 | 44.3 | 54.0 | 83.0 | 96.6 | 30.6 | 37.2 |
| | BGE-large-en-v1.5 | 335M | 35.7 | 52.8 | 60.9 | 67.2 | **88.5** | **98.7** | 45.4 | 49.2 |
| | BGE-m3 | 560M | 20.0 | 26.8 | 34.9 | 41.7 | 70.2 | 89.4 | 24.7 | 29.4 |
| Finetuned | MedCPT | 109M | 11.9 | 17.9 | 21.7 | 28.9 | 56.6 | 88.5 | 15.5 | 19.4 |
| | Pubmedbert-base-embeddings | 109M | 17.9 | 34.0 | 40.9 | 52.3 | 82.1 | 97.0 | 26.4 | 33.8 |
| | MedEmbed-small-v0.1 | 33.4M | 27.2 | 40.0 | 49.8 | 57.4 | 82.6 | 96.6 | 35.1 | 41.2 |
| | MedEmbed-base-v0.1 | 109M | 23.8 | 40.0 | 44.7 | 56.6 | 79.1 | 96.2 | 31.9 | 39 |
| | MedEmbed-large-v0.1 | 335M | 30.2 | 43.8 | 50.2 | 60.9 | 86.4 | 97.0 | 37.9 | 44.5 |
| | BMRetriever-410M | 410M | 31.9 | 46.4 | 53.6 | 61.3 | 80.9 | 96.6 | 39.9 | 45.8 |

Table 13: Percentage improvement in retrieval using a reranker (BGE-reranker-large).

| Model | Recall@10↑ | nDCG@10↑ |
|---|---|---|
| All-MiniLM-L12-v2 | -24.8 | -34.4 |
| E5-small-v2 | -26.1 | -36.7 |
| Contriever-msmarco | -33.6 | -44.1 |
| Contriever | -33.3 | -41.4 |
| BGE-small-en-v1.5 | -43.9 | -52.7 |
| BGE-base-en-v1.5 | -46.2 | -56.8 |
| BGE-large-en-v1.5 | -50.5 | -59.1 |
| BGE-m3 | -50.0 | -60.4 |
| MedCPT | -24.1 | -23.7 |
| Pubmedbert-base-embeddings | -43.7 | -50.3 |
| MedEmbed-small-v0.1 | -48.2 | -53.7 |
| MedEmbed-base-v0.1 | -57.0 | -60.4 |
| MedEmbed-large-v0.1 | -52.2 | -63.5 |
| BMRetriever-410M | -49.7 | -60.2 |

## A.7 ERROR MODES FOR RAG

Retrieval Success + RAG Incorrect is greater than Retrieval Fail + RAG Correct or Retrieval Fail + RAG Incorrect, suggesting that medical reasoning is a challenging task even when a correct document is provided. Models with good performance (GPT-5, Thinking Claude 4 Sonnet) have low Retrieval Success + RAG Incorrect, implying that pointing to the correct document often results in an accurate answer if the model's performance is good. Their high Retrieval Fail + RAG Correct also emphasizes the necessity of reasoning ability in answering challenging medical questions, which aligns with our findings in the paper.

The significance of the case reports are categorized into three types: treatment, test, and diagnosis. Correspondingly, OGCAREBENCH's questions are categorized into one of three significances. Figure 4a shows the distribution of significance over OGCAREBENCH. The dataset is dominated by treatment, whereas test and diagnosis form smaller proportions. This imbalance is natural in clinical case reports, as many of the novelties are within the boundary of treatment compared to a new disease or a detection method.

We conduct error analysis by investigating the significance of Retrieval Success + RAG Incorrect examples. Figure 4b shows the distribution of significance by models. Test and diagnosis questions' Retrieval Success + RAG Incorrect are similar throughout the models, and treatment makes most of the difference. This indicates good-performing models make difference during answering treatment-related questions. Similarity of test and diagnosis across multiple models also imply some questions may be naturally hard to answer. As GPT-5 and Thinking Claude 4 Sonnet have lower number of test and diagnosis counts, they show a positive performance gap through answering those questions correctly. This suggests that reasoning capabilities improve performance on more challenging questions.

Table 14: Retrieval and RAG performance breakdown for each model. BGE retriever with 3 context documents used. RS = Retrieval Success, RF = Retrieval Failure, RC = RAG Correct, RI = RAG Incorrect.

| Model | RAG Acc. | RS + RC | RF + RC | RS + RI | RF + RI |
|---|---|---|---|---|---|
| GPT-5 | 72.8 | 144 | 27 | 32 | 32 |
| GPT-o3-mini | 71.9 | 137 | 32 | 39 | 27 |
| Llama 3.3 70B Instruct | 66.0 | 131 | 24 | 45 | 35 |
| Claude 4 Sonnet | 67.7 | 135 | 24 | 41 | 35 |
| Thinking Claude 4 Sonnet | 73.6 | 144 | 29 | 32 | 30 |
| MedGemma-27b-text-it | 63.0 | 122 | 26 | 54 | 33 |
| Llama 3-Med42-70B | 56.2 | 111 | 21 | 65 | 38 |
| OpenBioLLM-Llama 3-70B | 48.1 | 94 | 19 | 82 | 40 |

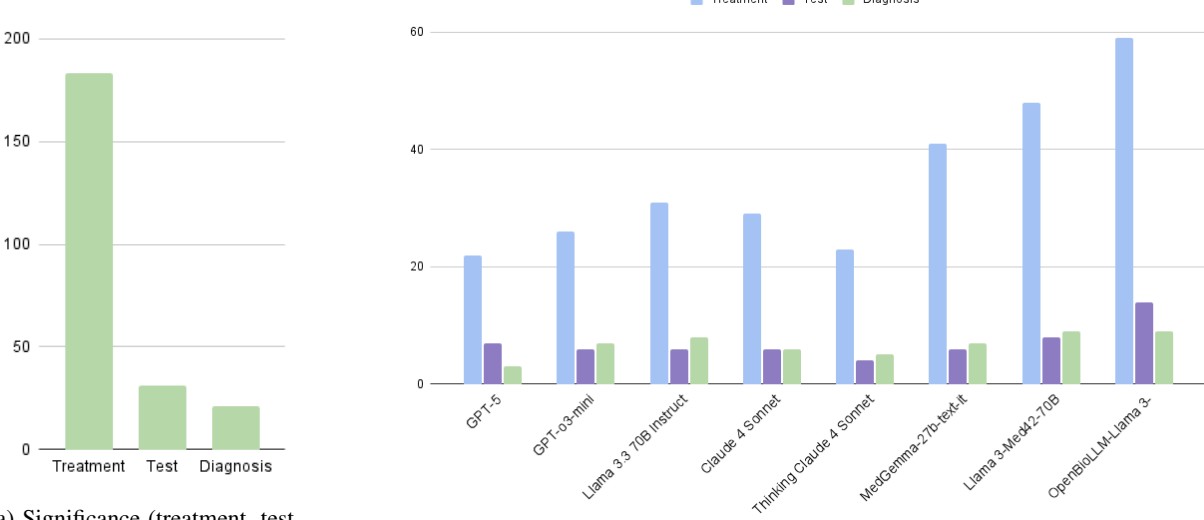

(a) Significance (treatment, test, diagnosis) distribution of the full dataset.

(b) Significance distribution of RAG incorrect despite retrieval success cases by model.

Figure 4: Distribution of significance for error analysis.

## B  CASE REPORT VENUES

| | | |
|---|---|---|
| Case Reports Plast Surg Hand Surg | Case Rep Cardiol | SAGE Open Med Case Rep |
| Int J Med Pharm Case Reports | JACC Case Rep | Case Rep Genet |
| J Clin Cases Rep | Case Rep Clin Med | Case Rep Dent |
| J Clin Case Rep | Case Rep Surg | Int Med Case Rep J |
| Case Rep Pancreat Cancer | Trauma Case Rep | Case Rep Ophthalmol |
| Arch Clin Case Rep | Case Rep Pulmonol | Autops Case Rep |
| Respir Med Case Rep | Clin Med Insights Case Rep | J Med Case Reports |
| Case Rep Neurol Med | Clin Pract Cases Emerg Med | Gynecol Oncol Case Rep |
| Case Rep Transplant | Case Rep Endocrinol | Clin Med Rev Case Rep |
| J Vasc Surg Cases Innov Tech | Med Case Rep Short Rev | Psychiatry Res Case Rep |
| Clin Nephrol Case Stud | Clin Case Rep Rev | Gen Thorac Cardiovasc Surg Cases |
| Arch Med Case Rep | Case Rep Emerg Med | MOJ Clin Med Case Rep |
| Endocrinol Diabetes Metab Case Rep | Prof Case Manag | IDCases |
| J Cardiol Case Reports | Ann Clin Case Rep | J Cardiol Cases |
| Am J Med Case Rep | Case Reports Immunol | Spinal Cord Ser Cases |
| Med Mycol Case Rep | Int Clin Med Case Rep J | Oxf Med Case Reports |
| Case Rep Psychiatry | IJU Case Rep | Case Rep Otolaryngol |
| J Surg Tech Case Rep | JCEM Case Rep | Case Rep Ophthalmol Med |
| Clin Med Case Rep | Case Rep Dermatol | JAAD Case Rep |
| Ann Clin Med Case Rep | J Pediatr Surg Case Rep | ACG Case Rep J |
| Surg Case Rep | Am J Ophthalmol Case Rep | Case Rep Crit Care |
| Case Rep Orthop | Case Rep Vet Med | Clin Case Stud |
| Case Rep Perinat Med | GMS Ophthalmol Cases | CASE (Phila) |
| Case Rep Radiol | Case Rep Womens Health | Eur Heart J Case Rep |
| Open J Clin Med Case Rep | Case Rep Gastrointest Med | Case Rep Infect Dis |
| Case Stud Eng Fail Anal | Case Rep Oncol Med | Cases J |
| BJR Case Rep | J Surg Case Rep | Case Stud Chem Environ Eng |
| Case Rep Dermatol Med | Clin Case Rep | Indian J Ophthalmol Case Rep |
| Case Rep Pediatr | BMJ Case Rep | Urol Case Rep |
| CEN Case Rep | Case Rep Anesthesiol | J Med Case Rep |
| Case Rep Urol | Case Reports Hepatol | Int J Surg Case Rep |
| Case Rep Obstet Gynecol | Int J Case Rep Imag | J Investig Med High Impact Case Rep |
| Asploro J Biomed Clin Case Rep | Case Rep Nephrol | Case Rep Hematol |
| Respirol Case Rep | Case Rep Pathol | Int J Clin Case Rep Rev |
| HeartRhythm Case Rep | Neurocase | AACE Clin Case Rep |
| Retin Cases Brief Rep | Cold Spring Harb Mol Case Stud | Case Stud Transp Policy |
| JBJS Case Connect | J Endourol Case Rep | Case Rep Rheumatol |
| Case Rep Med | Oral Health Case Rep | Arch Clin Med Case Rep |
| JMM Case Rep | Radiol Case Rep | Epilepsy Behav Case Rep |
| Case Rep Vasc Med | APSP J Case Rep | |

Table 15: Journal names of the corpus of 53,617 case reports. Extracted from PMC commercially available file list provided by: `https://ftp.ncbi.nlm.nih.gov/pub/pmc/oa_non_comm_use_pdf.csv`

## C    PROMPTS AND INSTRUCTIONS

**BMRetriever Instruction**

**Query template.** Given a question, retrieve relevant documents that best answer the question. *<query>*
**Passage template.** Represent this passage\n passage: *<passage>*

Figure 5: Instruction used by BMRetriever.

**GPT-4o prompt for significance extraction**

Please carefully read the provided case report text or abstract. Identify whether the report describes any unique clinical actions from the following list:

Novel treatment or drug introduced

Existing treatment used in a new way or indication

New surgical or procedural technique applied

Innovative combination of treatments or devices

Novel diagnostic test or imaging method used

Advanced molecular/genetic testing guiding treatment

Unique point-of-care or biomarker test employed

Use of AI or machine learning for diagnosis or treatment planning

Novel intervention to manage unexpected treatment complications

Off-label drug use with unique dosing or delivery method

Personalized or precision medicine approach in therapy

New integrated multidisciplinary care strategy

Innovative use of telemedicine or remote monitoring in clinical management

New rehabilitation or follow-up protocol applied

Novel preventive or screening intervention implemented

Ethical or legal decision-making impacting treatment

Use of newly developed medical devices or technologies for treatment

For each identified action, briefly summarize what it is and how it is unique or novel in this case report. If none apply, output exact string "no" (without quote marks) only. If there are multiple points, only describe the MOST SIGNIFICANT POINT, most likely mentioned in the abstract.

Figure 6: GPT-4o Prompt we used to extract significance from the case reports. The output is used to create the question-answer pairs.

**GPT-4o prompt for timeline extraction**

You will receive a clinical case presentation. Your task is to carefully parse and organize all clinical information strictly as a chronological timeline, listing events sequentially based on when they occurred or inferred from context. Include ALL significant events: arrival, initial presentation, interventions, medications administered, diagnostic tests and results, progression events, clinical decisions, and final diagnosis.

Patient Arrival: Infer when and how the patient initially presented.

Initial Clinical Presentation: Initial symptoms, vital signs, mental status, or general condition upon arrival.

Early Medications and Interventions: Any initial treatments or interventions (medications, fluids, oxygen, etc.) performed upon or soon after presentation.

Diagnostic Tests and Results: Tests ordered, procedures performed, and their outcomes. Infer order based on context if not explicitly stated.

Clinical Progression: Any changes in the patient's condition over time, including improvements, deteriorations, or significant clinical events.

Final Diagnosis or Impression: The final diagnosis, clinical assessment, or conclusion.

Avoid categorizing by types of events; instead, present them sequentially as they unfold in real clinical practice. Use exact sentences from the case report for building the timeline. DO NOT include any information that is not the a part of case presentation. Within the case presentation, include any information necessary for clinical decision making, such as the patient refusing or requireing certain diagnostic tools.

Output should be formatted in similiar way as this for parsing purpose:
- patient information
- initial symptoms and patient state
- following timeline in chronological order
...
- final timeline of the case report

Figure 7: GPT-4o Prompt we used to extract timeline from the case reports. The output is used to create the question-answer pairs.

**GPT-4o prompt for question-answer pair generation**

You are given a timeline and significant point of a medical case report. The significance refers to the new technique that the case report introduces. It can be surgical method, treatment, etc. Significance mentions the "new" item and explains why it is new. Timeline is an ordered list of bullet-style events exactly as they appear in the case report.

Create one question-answer pair asking for the immediate next step at the moment just before the action happens. The question should be the steps before taking the "new" action introduced in the significance. It should include all the steps before the decision point of "new" action. If there are multiple significant points in the input, choose the most complicated and unique one.

The question should not be based on an outcome but asking what is the significance or next step in treatment, or whether certain treatment/test could be done for this specific case, what to keep in mind during the treatment, etc. It should not ask about the past.

Extract a concise answer, using only exact sentence(s) from the timeline. The answer must be verbatim, not paraphrased.

This is a text only communication, so neither question nor answer should be referring to images, tables, or any other non-text media.

Output:

Context: Clearly list EVERY step of the timeline BEFORE this decision point in chronological order without leaving out any bulletpoints. ANY POINTS AFTER THE DECISION POINT SHOULD NOT BE IN THE CONTEXT. patient arrival, symptoms, vital signs, prior history, treatments/interventions, medications administered, diagnostic tests and their results, and clinical progression events—strictly up to this decision point should be included. If two consecutive elements of timeline are "decided to do xxx" and "xxx was performed", state it only once.

Question: Given this information, what is the immediate next appropriate clinical step? (Note: Adapt this to fit the situation, e.g., "next appropriate test", "next appropriate treatment", etc.)

Answer: State the exact next clinical step taken, using precise procedural or diagnostic terminology. Only use information explicitly stated in the timeline. THIS SHOULD NOT BE INCLUDED IN THE END OF THE CONTEXT. THIS IS THE **NEXT** STEP AFTER THE QUESTION STATEMENT.

The response should be outputted in json format:

```
{
  "Context": "contents",
  "Question": "contents",
  "Answer": "contents"
}
```

Figure 8: GPT-4o Prompt we used to create question-answer pairs from the case reports. Output "Context" and "Question" are combined to form the question.

**Claude 4 Opus prompt for controlled question modification**

You are given three items: (1) a detailed clinical query, (2) its corresponding concise query, (3) the answer.

Your task is to rewrite the **detailed query** following the instructions below:

Instructions:
1. Identify all overlapping content between the detailed and concise queries.
2. You must **preserve** the meaning of all overlapping content exactly yet **modify** the words or expressions.
- Keep the core meaning **unchanged**, but vary the surface form:
- Use synonyms, abbreviations, or different phrasing.
- Do NOT alter the medical intent or expected answer.
- Example: "Management of acute MI" → "Initial treatment of a heart attack".
3. Identify the non-overlapping parts of the detailed query:
- Use synonyms, abbreviations, or different phrasing.
- Adjust numerical values by adding or subtracting within medically reasonable ranges.
- Altering the logical flow, sentence structure, or clinical context.
4. Add **extra distracting medical content** that are medically plausible but irrelevant to the answer:
- Comorbidities - Symptoms, tests, and treatments.
- Background information.
- Past but resolved medical history.
- Family history that does not affect the answer.
- Redundant or vague phrases.

**IMPORTANT:**
- The revised detailed query should look **substantially different** from the original, while remaining **medically plausible**.
- It must still be **answerable by the original answer**.
- Return **only the modified detailed query**.

Figure 9: Claude 4 Opus prompt we used to "distract" questions. The detailed question is the full question from Figure 8 output. The concise question is the part that is necessary to derive the answer, while a detailed question usually conveys details unnecessary to reach the answer. Concise question remains semantically unchanged while other details are notably modified.

**(a) General-purpose model prompt**

You are a helpful medical assistant answering expert-level medical questions. You will receive a detailed clinical question about a patient case. Answer the question with one best answer. Do not generate multiple answers.

(a) Prompt for general-purpose models.

**(b) Medical domain model prompt**

You are a helpful medical assistant answering expert-level medical questions. You will receive a detailed clinical question about a patient case. Answer the question with one best answer. Do not generate multiple answers. Do not include analysis, steps, or thoughts, and restrict response to less than 100 words.

(b) Prompt with content and length restriction, for medical domain models

Figure 10: Prompts for generating responses with the questions from OGCAREBENCH.

**GPT-4o prompt for evaluating answer and model response equivalence**

You are given two texts: a gold standard answer and a response. Both describe the next step in medical procedure for a specific patient. Your task is to determine whether the response conveys the same medical intent as the gold answer.

Follow these steps:

Identify the core medical action(s) in the gold answer. Express them as concise medical actions (e.g., "start chemotherapy," "perform lobectomy," "order CT scan"). Ignore extra details like dose, frequency, or technique unless they fundamentally change the type of action.

Identify the core medical action(s) in the response. Use the same criteria.

Compare the two sets of actions:

Consider them equivalent if they refer to the same kind of medical action, even if wording differs. If the main medical procedure is similar and other details somewhat aligns, two texts are equivalent.

Mark as mismatch if the response suggests a different type or intent of medical procedure. If two texts include similar medical procedure but their importance differs greatly, or their main medical procedure differs, mark as mismatch.

Pay special attention to whether the response changes the stage or intent of medical procedure (e.g., monitoring vs. intervention, surgery vs. medication). This counts as mismatch.

Some texts have reasons or rationale explaining their main content. Do not use this part to determine equivalence or mismatch.

Write your evaluation in plain text as:

Equivalence if the response implies the same medical intent as the gold answer.

Mismatch if the response implies a different medical intent.

The output format should be: "Equivalent" or "Mismatch". Do not output any other texts.

Few-shot examples
[Omitted for brevity]

Figure 11: GPT-4o prompt we used to evaluate the answer and LLM equivalency. The few-shot examples are drawn from data that was excluded from the final dataset but received relatively high ratings (scored 3 out of 5).

**Instruction for the annotators**

There are 7 columns in the spreadsheet:

Title: title of the case report where the query and answer were derived from.

Classification: the topic of the case report.

Link: link to the case report. If you have any confusion or want to review the case report for validation, please use this link. If the query is straightforward, you don't need to validate with the case report.

Query: question presenting the case that is similar to the case report and asking the next step at a potentially confusing decision point, related to the significant point, or the reason why the case report was written. The query will not be a direct iteration of the case report, as we added distractions to make the query more interesting and difficult to answer.

Answer: answer to the query, derived from the case report.

Rating: your rating of the query-answer pair, in 1 to 5 scale.

Comments: your comments about the query-answer pair. Feel free to leave it blank. You do not have to justify your rating or write detailed comments.

We removed the GPT responses to avoid potential hallucinations they may cause on the annotation.

Rating and Comments columns are for you to fill with your opinions. We want to simulate a situation where a doctor is looking up a case report for reference, and whether the action will be taken or not is up to the doctor. Our goal is to test if the models are reliable for that purpose using our dataset. The gold answers on the spreadsheet reflect the answer presented by the case report, whether it is the standard or not. We want to confirm that our information extraction was successful and that our queries are not too easy or obvious and correct (okay if the answer is not a common action).

To serve these purposes, the rating should be based on these criteria:

- The answer should be answering the query.
- The query should not be asking too "easy" question. The query should require knowledge from medical professionals to be answered.
- There are some hallucinated data (e.g., answer is already stated on the question). We did our best to filter them using a model, but if you see malformed data, feel free to rate it as 1 without further consideration.

These are okay:

- The answer is not gold standard (The goal is to create a dataset reflecting the case report's action taken.)
- The answer is not detailed
- The answer is too specific to the patient (Some information, such as numbers, may be too specific to the patient, but the doctor potentially searching this case will also see the same information.)

Figure 12: Instruction given to three annotators to verify question-answer pairs.

