# OpenReview forum: "When Case Gets Rare: A Retrieval Benchmark for Off-Guideline Medical Question Answering"
_ICLR.cc/2026/Conference — Submitted to ICLR 2026_

### Official Review · Reviewer_ASoa · 2025-10-24

**Soundness:** 2
**Presentation:** 3
**Contribution:** 2
**Rating:** 2
**Confidence:** 5

**Summary:**

This paper presents a new benchmark and dataset derived from published case reports. The benchmark is filtered by physicians to ensure validity and relevance to the task defined by the authors. The resulting evaluation is then used to obtain a baseline for a variety of models including closed and open models. The authors then augment the models by using RAG over the case reports dataset. Different embedding models are used and compared. Overall, this work shows that adding RAG improves the performance on the OGCAREBENCH evaluation compared to the models without RAG.

**Strengths:**

The dataset curated by the authors has merit and the question generation pipeline is robust. The experimental results are clear and demonstrate the relevance of RAG for improving performance on under represented data such as rare medical cases. Including multiple embedding models supports the need for adapted models and suggest that focusing on improving RAG may be a more viable approach than training LLMs for domain specialization.

**Weaknesses:**

The assumptions made by the authors on what the goals of care are and how clinicians practice medicine are questionable. Rare cases are by definition rare and often have non-specific presentations shared with more common diseases [1]. The assumption that the best next step for a known rare case is the best next step in a real-world encounter is incorrect. For example, if a 20 year old male presents to the ER with shortness of breath and imaging shows a pneumothorax, the optimal diagnostic step is not to look for FLCN genetic mutations whereas that would be the case if in a Birt Hogg Dube case report [2].

This misalignment raises questions about the applicability of a system focusing on rare cases. For rare kidney diseases, a biopsy is the gold standard to make the diagnosis [3], would using this system in real world settings suggest a biopsy for every patient presenting with proteinuria or altered kidney function? Does the system rely on the clinician's intuition of what is or isn't a rare case? What are the cost and outcome implications of a system proposing advanced interventions too quickly?

This misalignment adds to the existing disconnect between lab research and the reality of clinical practice. While the experience is interesting, I do not find the method, system, and results to be novel enough to compensate the lack of real-world relevance and insufficient framing as is.

Considering the baseline and comparison, having a baseline of classic inference is necessary but insufficient, to justify using this RAG approach, it should be compared to other RAG systems, GPT-5 with web search for example, deepsearch, and OpenEvidence would be more accurate comparisons.

The inclusion of OpenBioLLM is a major issue for me [4]. A model released without any information, paper or data description making bold claims of achieving SOTA should not be included in any scientific work especially given the results reported in this paper.

# References

[1] Rare inherited kidney diseases: challenges, opportunities, and perspectives (Devuyst et al. 2014)

[2] Birt-Hogg-Dube Syndrome (Crane et al. 2023)

[3] The Kidney, (Brenner., and Rector. 2019)

[4] aaditya/Llama3-OpenBioLLM-70B (2024)

**Questions:**

# Suggestions/Questions

1) Include in the baseline the performance of base models with web search usage (GPT-5 Search, Claude with web use, deepsearch).

2) I recommend removing OpenBioLLM from the paper and in general to avoid models from unknown sources without data/technical report.

3) Evaluate the performance of the system on common cases, does it make rare suggestions? Could you quantify the incidence of the cases in the evaluation dataset and quantify the pre- and post-test likelihood of the correct answer? In addition, I would like to see a benefit/risk assessment beyond accuracy. For example, missing causa equina syndrome has more impact than missing a birt hogg dube diagnosis. Likewise, performing a kidney biopsy on a healthy patient is more risky than a blood sample. Finding the optimal benefit/risk ratio is the primary objective in clinical practice, not accuracy [1].

4) An expert baseline on the benchmark would help put into perspective the performance of the system.

5) Was any method used to ensure the absence of contamination between the evaluation cases and the cases included in the RAG dataset?

# References

[1]  Comparing diagnostic tests on benefit-risk (Pennello et al., 2016)

---

> ### Author Response · Authors · 2025-11-21
>
> Thank you for your clinically rich comments. As we are focusing on an expert domain, it is essential to consider the domain characteristics. We have a long response, so we will split it into multiple comments. References will be in the last comment.
>
> We would like to clarify that our contribution is a benchmark containing instances of rare cases, not a model or system to solve rare cases. Performance on common cases is therefore not applicable in our case.
>
> >The assumptions made by the authors on what the goals of care are and how clinicians practice medicine are questionable. Rare cases are by definition rare and often have non-specific presentations shared with more common diseases [1]. The assumption that the best next step for a known rare case is the best next step in a real-world encounter is incorrect. For example, if a 20 year old male presents to the ER with shortness of breath and imaging shows a pneumothorax, the optimal diagnostic step is not to look for FLCN genetic mutations whereas that would be the case if in a Birt Hogg Dube case report [2].
>
> The benchmark is for real-world rare cases rather than real-world common cases. As we create the benchmark directly from the case reports, we won’t encounter such cases. We are expecting physicians would have taken appropriate measures to treat common symptoms or diseases and refer to case reports when they feel consulting is necessary.
>
> >This misalignment adds to the existing disconnect between lab research and the reality of clinical practice. While the experience is interesting, I do not find the method, system, and results to be novel enough to compensate the lack of real-world relevance and insufficient framing as is.
>
> Thank you for highlighting the domain specific aspects. Our goal is to introduce a controlled and reproducible task to evaluate LLM performance in the medical domain that is beyond general knowledge and the official guidelines, rather than to evaluate clinical procedures directly. We want to introduce a new challenge in the NLP community in terms of domain-specific tasks. As direct evaluation of models in real-world clinical procedures is infeasible and cannot be easily reproduced, our benchmark is intended to serve as a proxy measure to understand if an LLM can solve/help clinicians solve rare medical cases where high-level medical reasoning is necessary.
>
> The empirical results show the gap that was not previously identified. We identify the shortcomings of SOTA LLMs in previously unseen medical questions requiring deep reasoning unless supported by retrieval, even though their performance against other medical benchmarks is reported to be good.
>
> >Considering the baseline and comparison, having a baseline of classic inference is necessary but insufficient, to justify using this RAG approach, it should be compared to other RAG systems, GPT-5 with web search for example, deepsearch, and OpenEvidence would be more accurate comparisons.
>
> >Include in the baseline the performance of base models with web search usage (GPT-5 Search, Claude with web use, deepsearch).
>
> We included a web search enabled model, GPT-4o-search-preview, as our baseline (Table 4, Line 363). This was the most recent model available via an API at the time of submission.
>
>
> >The inclusion of OpenBioLLM is a major issue for me [4]. A model released without any information, paper or data description making bold claims of achieving SOTA should not be included in any scientific work especially given the results reported in this paper.
>
> >I recommend removing OpenBioLLM from the paper and in general to avoid models from unknown sources without data/technical report.
>
> We included OpenBioLLM as it was used in other papers[1, 2] and was made by Saama AI, with an acknowledgement from Meta [3]. Thank you for pointing out the issues with this model, we will include a disclaimer in the updated draft.
>
>
> >Could you quantify the incidence of the cases in the evaluation dataset and quantify the pre- and post-test likelihood of the correct answer? In addition, I would like to see a benefit/risk assessment beyond accuracy. For example, missing causa equina syndrome has more impact than missing a birt hogg dube diagnosis. Likewise, performing a kidney biopsy on a healthy patient is more risky than a blood sample. Finding the optimal benefit/risk ratio is the primary objective in clinical practice, not accuracy [1].
>
> Thank you for bringing this up. For the benefit and risk, we are assessing if the model can answer rare case clinical questions. Our goal is to simulate a situation where the doctor is consulting with the model instead of going through the case reports through traditional search. The final clinical decision is up to the physician.
>
> For quantifying pre- and post-test likelihood of the correct answer, could you provide clarification or a suggestion of what we could do?

---

> ### Author Response · Authors · 2025-11-21
>
> >An expert baseline on the benchmark would help put into perspective the performance of the system.
>
> Thank you for raising this. However, an expert baseline is hard and time-consuming, therefore not done by many benchmarks. For example, MedicationQA [4] doesn't have an expert baseline but has an expert validating the answers, like ours. HealthSearchQA [5] has expert answers to their QA pairs for only a subset, which is not publicly available.
>
> >Was any method used to ensure the absence of contamination between the evaluation cases and the cases included in the RAG dataset?
>
> We apply significant modifications to the raw question extracted from the source case report to ensure that we do not have a circular task design of retrieving what we extracted. The modifications include altering patient demographics (e.g., age and ethnicity), substituting medical terminology with semantically equivalent expressions, and integrating comorbidities that do not affect the original condition, etc. Modified questions are our end product and thus annotated by physicians. This process was to make the task realistic and suitable for RAG. If we are misinterpreting your question, please let us know.
>
>
> **References**
>
> [1] Shoham, Ofir Ben, and Nadav Rappoport. "MedConceptsQA: Open source medical concepts QA benchmark." Computers in Biology and Medicine 182 (2024): 109089.
>
> [2] Dorfner, Felix J., et al. "Biomedical large languages models seem not to be superior to generalist models on unseen medical data." arXiv preprint arXiv:2408.13833 (2024).
>
> [3] “How Llama Is Helping Saama Deliver New Possibilities in Personalized Medicine and Data-Driven Care.” Meta.com, 2022, ai.meta.com/blog/saama-data-driven-care-built-with-llama/.
>
> [4] Abacha, Asma Ben et al. “Bridging the Gap Between Consumers' Medication Questions and Trusted Answers.” Studies in health technology and informatics vol. 264 (2019): 25-29. doi:10.3233/SHTI190176
>
> [5] Singhal, Karan, et al. "Large language models encode clinical knowledge." arXiv preprint arXiv:2212.13138 (2022).

---

> > ### Author Response · Authors · 2025-11-26
> >
> > **Dear Reviewer ASoa, Thank you again for your review of our work. We understand you might be busy, but we hope you will be able to review our rebuttal. We are happy to clarify or elaborate on any points as needed. If our response addresses your concerns and positively influences your view of the work, we would be grateful if you reflect that in an updated score.**

---

> > > ### Comment · Reviewer_ASoa · 2025-11-27
> > >
> > > Dear authors,
> > >
> > > My assessment of the work remains unchanged after the response. Please find my comments/questions based on your response.
> > >
> > > > We are expecting physicians would have taken appropriate measures to treat common symptoms or diseases and refer to case reports when they feel consulting is necessary.
> > >
> > > This is precisely the challenge with rare cases: they often present with non-specific symptoms (such as fatigue) that look like common diseases. The difficulty in clinical settings is deciding if a case would benefit from further investigations or orientation towards a specialist. Once orientation to the correct specialist is made, diagnosis is usually quick and straightforward.
> > >
> > > > The empirical results show the gap that was not previously identified. We identify the shortcomings of SOTA LLMs in previously unseen medical questions requiring deep reasoning unless supported by retrieval, even though their performance against other medical benchmarks is reported to be good.
> > >
> > > The challenge with this affirmation is that offguidelines decisions may not be optimal. Case reports are usually a narrative description of what happened and what the team did, which may or may not have been the gold standard of care. Relying on these cases as a gold standard and claiming that LLMs demonstrate low performance may introduce a bias, whereas using RAG would just demonstrate that LLMs are steered towards the clinical realities, not necessarily the optimal care.
> > >
> > > > For quantifying pre- and post-test likelihood of the correct answer, could you provide clarification or a suggestion of what we could do?
> > >
> > > For diagnosis and best next step questions, reporting the incidence of the case and the LR+ of the suggested test would allow us to understand if the tool has a high chance of generating false positives and, therefore, gauge the potential harm of using this tool in a production environment.
> > >
> > > > Thank you for raising this. However, an expert baseline is hard and time-consuming, therefore not done by many benchmarks. For example, MedicationQA [4] doesn't have an expert baseline but has an expert validating the answers, like ours. HealthSearchQA [5] has expert answers to their QA pairs for only a subset, which is not publicly available.
> > >
> > > While I agree that getting a human baseline is challenging, I believe it to be a necessity and that previous work, which did not include such baselines, should be seen as a historical issue that we should collectively move away from. Applied ML to clinical problems, should keep in mind that the ultimate goal is to help patients, healthcare workers must be involved to understand the real-world potential of these tools and provide an interpretable baseline of what would be the minimal performance target to achieve.
> > >
> > > > We apply significant modifications to the raw question extracted from the source case report to ensure that we do not have a circular task design of retrieving what we extracted. The modifications include altering patient demographics (e.g., age and ethnicity), substituting medical terminology with semantically equivalent expressions, and integrating comorbidities that do not affect the original condition, etc. Modified questions are our end product and thus annotated by physicians. This process was to make the task realistic and suitable for RAG. If we are misinterpreting your question, please let us know.
> > >
> > > My question was related to contamination, transforming the original case text into a question, but retaining the case itself in the RAG database would still be considered contamination. Are cases included in the RAG database a different subset from the cases used to generate questions?

---

> > > > ### Author Response · Authors · 2025-12-03
> > > >
> > > > Thank you for your comments.
> > > >
> > > > >This is precisely the challenge with rare cases: they often present with non-specific symptoms (such as fatigue) that look like common diseases. The difficulty in clinical settings is deciding if a case would benefit from further investigations or orientation towards a specialist. Once orientation to the correct specialist is made, diagnosis is usually quick and straightforward.
> > > >
> > > > We appreciate the clarification. Our benchmark is not designed to test the kind of cases you point out to make a determination of whether a case needs to be seen by a specialist. Once a case is oriented to the correct specialist, we are trying to emphasize that those specialists consult with external resources when they encounter an unusual case. This step will occur after the orientation to the correct specialist and **by the specialist**. We interviewed 10 physicians from different specialties and confirmed that this is indeed the case (line 135-140). Aligning with your comment, emergency medicine practitioners do not consult with case reports, while surgery and internal medicine (our two major focuses in the benchmark) practitioners do. We focus on the stage where clinicians have recognized the limits of the standard guidelines and therefore consult case reports. Our contribution is to indicate the limitations of existing models’ ability to assist physicians under these scenarios using our benchmark dataset.
> > > >
> > > > >The challenge with this affirmation is that offguidelines decisions may not be optimal. Case reports are usually a narrative description of what happened and what the team did, which may or may not have been the gold standard of care. Relying on these cases as a gold standard and claiming that LLMs demonstrate low performance may introduce a bias, whereas using RAG would just demonstrate that LLMs are steered towards the clinical realities, not necessarily the optimal care.
> > > >
> > > > Thank you for the comment. We are not treating case reports as the standard of care, but testing LLMs’ capabilities to provide suggestions to physicians by retrieving case reports. The decision is always up to the physicians, as in a situation where the physician manually searches or refers to case reports. RAG does not steer models toward suboptimal care but rather aims to assist medical reasoning by providing a context to refer to. Therefore, our findings reflect limitations in models’ ability to use clinical context rather than a claim about optimal care.
> > > >
> > > > >For diagnosis and best next step questions, reporting the incidence of the case and the LR+ of the suggested test would allow us to understand if the tool has a high chance of generating false positives and, therefore, gauge the potential harm of using this tool in a production environment.
> > > >
> > > > We appreciate the suggestion, but our contribution is a dataset and not a model, and therefore, this is not applicable to our contribution.
> > > >
> > > > >My question was related to contamination, transforming the original case text into a question, but retaining the case itself in the RAG database would still be considered contamination. Are cases included in the RAG database a different subset from the cases used to generate questions?
> > > >
> > > > Thank you for raising this concern. We would like to emphasize that our questions went through significant controlled modification, and a subset of questions before and after modification was reviewed by physicians to confirm that they look significantly different but result in the same answer. Our low retrieval success recall@1 also confirms this. **In a clinical setting, physicians consult with the existing corpus of case reports to resolve similar but not identical cases.** Our benchmark simulates this by retrieving real case reports while ensuring the questions are not a repetition of the source. Models can be evaluated as the questions in the dataset are from already documented cases. If we use an external source, such as ongoing cases, we cannot confirm the answer.

---

### Official Review · Reviewer_uPgu · 2025-10-27

**Soundness:** 3
**Presentation:** 3
**Contribution:** 2
**Rating:** 4
**Confidence:** 3

**Summary:**

This paper introduces OGCAREBENCH, a benchmark designed to evaluate large language models (LLMs) on rare or off-guideline medical cases. The benchmark is constructed from over 53,000 open-access medical case reports and includes 235 curated question–answer pairs validated by physicians. The dataset focuses on long-form, retrieval-based question answering where guideline-based reasoning is insufficient. Experiments compare several general-purpose and medical LLMs—both with and without retrieval augmentation (RAG)—demonstrating that even advanced models like GPT-5 struggle without retrieval (≈45–50% accuracy) but achieve substantial gains (up to 75%) when augmented with relevant case report retrieval. The paper argues that reliable clinical LLMs must move beyond memorized knowledge and toward retrieval-grounded reasoning to handle real-world, rare patient scenarios.

**Strengths:**

**Novel focus on off-guideline cases**:
The paper highlights an important but underexplored problem—how LLMs perform when faced with rare, atypical medical scenarios not covered by standard clinical guidelines. This makes the benchmark highly relevant to the deployment of LLMs in clinical support settings.

**Robust dataset design and validation**:
The authors construct OGCAREBENCH using a clear multi-step pipeline involving filtering, controlled question modifications, and expert validation. The attention to medical plausibility and domain fidelity adds credibility to the dataset’s reliability.

**Comprehensive experimental evaluation**:
The paper thoroughly evaluates multiple general-purpose and domain-specific models under both baseline and retrieval-augmented conditions. The inclusion of 15 retrieval models, from BM25 to biomedical-specific retrievers, offers a valuable empirical contribution to RAG research in healthcare.

**Clear demonstration of retrieval importance**:
Results convincingly show that RAG significantly enhances reasoning accuracy for rare medical cases, underscoring a key insight: parametric memory alone is insufficient for safe and effective medical reasoning.

**Weaknesses:**

**Limited methodological novelty**:
The work primarily focuses on dataset creation and empirical benchmarking rather than proposing a new retrieval or reasoning framework. This makes it somewhat engineering-heavy and evaluation-oriented, which may not align well with ICLR’s focus on algorithmic or representational innovation.

**Scale and representativeness concerns**:
Despite using over 50,000 case reports as the retrieval corpus, the final benchmark contains only 235 validated instances. This relatively small size raises questions about statistical robustness and whether the benchmark adequately covers the diversity of real-world rare cases.

**Evaluation dependency on GPT-based judging**:
The reliance on GPT-4o as an automatic evaluator introduces potential bias and inconsistency in clinical correctness judgments. The limited physician cross-validation (45 samples) may not be sufficient to confirm reliability across all cases.

**Shallow analysis of model failure modes**:
While quantitative results are extensive, the paper lacks qualitative insights into why models fail on certain rare cases—whether due to retrieval errors, reasoning gaps, or hallucinated procedures. This limits the interpretability of the findings.

**Questions:**

**Venue suitability (ICLR relevance)**:
The contribution centers on dataset construction and empirical evaluation, not on learning mechanisms or model training. It may better fit NLP or medical informatics venues (e.g., ACL, EMNLP) rather than ICLR, which emphasizes theoretical and representational advances.

**Benchmark longevity and updateability**:
Since medical knowledge evolves rapidly, the benchmark may require frequent updates to remain relevant. The paper does not discuss mechanisms for versioning, reannotation, or handling outdated clinical knowledge.

**Clinical validation and real-world utility**:
While the dataset is well-validated at construction time (Section 3.1 Step 1~4), it remains unclear how the benchmark correlates with actual clinical decision-making outcomes. Without external validation in real medical workflows, practical impact remains speculative. Furthermore, the authors did not provide any information about the annotators (i.e., three physicians).

---

> ### Author Response · Authors · 2025-11-21
>
> Thank you for your considerate reviews and questions! We have a long response, so we will split it into multiple comments. References will be in the last comment.
>
> >Limited methodological novelty: The work primarily focuses on dataset creation and empirical benchmarking rather than proposing a new retrieval or reasoning framework. This makes it somewhat engineering-heavy and evaluation-oriented, which may not align well with ICLR’s focus on algorithmic or representational innovation.
>
> >Venue suitability (ICLR relevance): The contribution centers on dataset construction and empirical evaluation, not on learning mechanisms or model training. It may better fit NLP or medical informatics venues (e.g., ACL, EMNLP) rather than ICLR, which emphasizes theoretical and representational advances.
>
> Dataset and Benchmark is listed as one of the topics in ICLR 2026 call for papers; this paper is well within the scope of this conference. Our novelty is introducing a new task of answering open-ended off-guideline rare-case questions with a non-trivially created benchmark.
>
>
> >Scale and representativeness concerns: Despite using over 50,000 case reports as the retrieval corpus, the final benchmark contains only 235 validated instances. This relatively small size raises questions about statistical robustness and whether the benchmark adequately covers the diversity of real-world rare cases.
>
> Benchmarks of this size are not unusual. Within the same domain, PubMedQA [1] has 500 expert-verified test samples, and  MedAESQA [2] has 40 examples. Outside of medical domain, MT-Bench [3] has 80 multi-turn questions, with each question having 2 turns.
>
> Many renowned benchmarks in the medical domain are often multiple-choice with expert-level questions or open-ended with non-expert-level questions, as mentioned in Section 2. While our benchmark seems small, it requires greater effort to verify as it is based on rare and novel cases, requiring experienced physicians to investigate each case reports carefully to fully validate the question-answer pair.
>
> We did a statistical analysis with our dataset, which is number 1 in the general comment (Statistical analysis for dataset size). According to its result, our benchmark of 235 QA pairs provides a stable performance estimate. While confidence intervals are relatively broad, we are able to capture the performance difference between the models. Also, the bootstrap mean being nearly identical to the reported accuracy indicates our benchmark is stable for model comparison.
>
> >Evaluation dependency on GPT-based judging: The reliance on GPT-4o as an automatic evaluator introduces potential bias and inconsistency in clinical correctness judgments. The limited physician cross-validation (45 samples) may not be sufficient to confirm reliability across all cases.
>
> Thank you for raising this. Using the Wilson confidence interval, the 95% confidence interval is 82.1%-97.7%. This surpasses the agreement levels in other popular benchmarks like MT-Bench [3], which has human-human agreement of approximately 81% and LLM-human agreement of above 80%.
>
> >Shallow analysis of model failure modes: While quantitative results are extensive, the paper lacks qualitative insights into why models fail on certain rare cases—whether due to retrieval errors, reasoning gaps, or hallucinated procedures. This limits the interpretability of the findings.
>
> Thank you for bringing this up. For failure analysis, we created a contingency table of retrieval and RAG. Please refer to number 3 of the general comment (Contingency table of retrieval and RAG) for the table and analysis. We aim to further look into retrieval success yet having incorrect RAG responses, focusing on criteria such as specialty or the significance type of case report (treatment, diagnosis, test).
>
> >Benchmark longevity and updateability: Since medical knowledge evolves rapidly, the benchmark may require frequent updates to remain relevant. The paper does not discuss mechanisms for versioning, reannotation, or handling outdated clinical knowledge.
>
> Thank you for raising this reasonable concern. Upon acceptance, we plan to publicize the full code and prompts used for the generation of our benchmark so that it can be updated as knowledge drifts over time. The pipeline is not hard to reproduce, and PMC allows mass public download of their papers via FTP. However, outdating of the dataset is not unique to our benchmark, and rather an issue for most of the benchmarks. This is also acknowledged in the limitations section.

---

> > ### Author Response · Authors · 2025-11-21
> >
> > > Clinical validation and real-world utility: While the dataset is well-validated at construction time (Section 3.1 Step 1~4), it remains unclear how the benchmark correlates with actual clinical decision-making outcomes. Without external validation in real medical workflows, practical impact remains speculative. Furthermore, the authors did not provide any information about the annotators (i.e., three physicians).
> >
> > Thank you for your considerate concern. The annotators are clinically active, licensed physicians in internal medicine and oncology. We are aiming to introduce a new task of answering open-ended questions about rare cases. We validated the practicality of this task through interviews with physicians. While this standalone cannot be directly applied in real medical workflows, we believe this is a necessary task in terms of the domain applicability of LLMs and testing clinical decision-making.
> >
> > **References**
> >
> > [1]Jin, Qiao, et al. "Pubmedqa: A dataset for biomedical research question answering." Proceedings of the 2019 conference on empirical methods in natural language processing and the 9th international joint conference on natural language processing (EMNLP-IJCNLP). 2019.
> >
> > [2]​​Gupta, Deepak, et al. “A Dataset of Medical Questions Paired with Automatically Generated Answers and Evidence-supported References.” Scientific Data, vol. 12, 2025, article no. 1035. Nature Publishing Group, https://doi.org/10.1038/s41597-025-05233-z
> >
> > [3] Zheng, Lianmin, et al. "Judging llm-as-a-judge with mt-bench and chatbot arena." Advances in neural information processing systems 36 (2023): 46595-46623.

---

> > > ### Author Response · Authors · 2025-11-26
> > >
> > > **Dear Reviewer uPgu, Thank you again for your review of our work. We understand you might be busy, but we hope you will be able to review our rebuttal. We are happy to clarify or elaborate on any points as needed. If our response addresses your concerns and positively influences your view of the work, we would be grateful if you reflect that in an updated score.**

---

### Official Review · Reviewer_wxnk · 2025-11-01

**Soundness:** 2
**Presentation:** 2
**Contribution:** 2
**Rating:** 2
**Confidence:** 5

**Summary:**

**Problem & Motivation:**
The paper identifies a gap between the training of most medical LLMs and the demands of real-world clinical practice. Current medical LLMs are primarily trained on and evaluated against common, "guideline-focused" medical knowledge, often using multiple-choice question formats. This reliance on parametric memorization is insufficient for the "long tail" of clinical care, where physicians encounter rare or "off-guideline" cases not covered by standard pathways. In these scenarios, evidence-based reasoning, which requires dynamically consulting external sources (like case reports), is essential. The authors argue that current benchmarks fail to test this critical skill, rarely evaluating whether models can generate expert-level, long-form answers grounded in retrieved evidence for complex, rare cases.

**Method:**
To address this gap, the paper introduces OGCAREBENCH, a long-form, retrieval-focused benchmark designed to evaluate LLMs on realistic, "off-guideline" clinical questions derived from medical case reports. The dataset's creation involves a four-step, semi-automatic process:
- *Corpus & Case Filtering:* A retrieval corpus of 53,617 case reports was collected from PubMed Central. From this, a subset of 28,219 reports was filtered by excluding older reports (published ≤ 2022) and those with high citation counts (indicating the case may have become standard knowledge).
- *Q&A Extraction:* An LLM (GPT-4o) was used on a 1,100-report subset to extract a timeline of the case and its significant contribution (e.g., a novel diagnosis, rare treatment). A question-answer pair was then generated, with the question detailing the case up to the decision point and the answer being the contribution (the novel step taken).
- *Question Modification:* To simulate a realistic clinical scenario where a new patient resembles but is not identical to a published case, the extracted questions were modified by another LLM (Claude 4 Opus). This involved adding "distractors" like altered demographics, comorbidities, or synonymous medical terms, making the question distinct from the source text.
- *Physician Validation:* The final modified Q&A pairs were validated by physicians, who rated them on medical alignment and difficulty. Only pairs requiring expert-level knowledge (rated 4 or 5 out of 5) were retained, resulting in the final benchmark of 235 validated cases.

**Experimental Setup:**
The benchmark is evaluated in two settings:
- Baseline (Memorization): Models answer questions using only their parametric knowledge.
- RAG (Retrieval-Augmented): Models are provided with relevant case reports retrieved from the 53k corpus to ground their answers.
A mix of general-purpose (e.g., GPT-5, GPT-03-mini, Llama 3.3) and medical-specific (e.g., MedGemma, Llama 3-Med42) models were tested.

**Results & Findings:**
The results demonstrate the benchmark's effectiveness in highlighting the limitations of memorization.
Without retrieval, even the best-performing model (GPT-03-mini) answered only 51.5% of questions correctly. Open-source models were lower, with MedGemma at 36.2%. This confirms that current models have not memorized this long-tail, rare-case knowledge.
When augmented with retrieved case reports, performance increased significantly. The top-performing model (GPT-5 with RAG) achieved 75.3% accuracy.

**Strengths:**

- The task of answering rare, off-guideline clinical questions is highly relevant.
- Timeline extraction and question reformulation by presenting all procedures preceding the decision point.
- A broad spectrum of models—domain-specialized and not—is benchmarked, including 8 LLMs, 14 semantic retrievers, and BM25.

**Weaknesses:**

- *Limited methodological novelty.* The primary contribution is the use of case reports as a source, not the development of a novel benchmark methodology. The resource is constructed by filtering and sampling (with poor control) an existing PubMed Central corpus, applying simple semi-automatic LLM-based extraction, and performing (incomplete) manual verification.
- *Superficial comparison to existing work.* For a resource-centric paper, the comparison to the existing literature is insufficient. It lacks a detailed, quantitative comparison table positioning OGCAREBENCH against other benchmarks (especially those already using case reports) across key dimensions (e.g., scale, task, validation rigor). This makes it difficult to assess the true novelty or "delta" provided by this work.
- *Questionable sampling and unbalanced dataset.*
   - Sampling. A "pure random sampling" of 1,100 reports (from which only 235 are finalized) is inadequate. Stratified sampling (e.g., by specialty, contribution type) would have been more rigorous.
   - Scale. The final expert-verified dataset of 235 questions is extremely small.
   - Balance. The dataset is highly unbalanced, with 70% of cases coming from only two specialties, a distribution that does not reflect the source corpus. The authors also fail to quantify the distribution of "contribution types" (diagnosis, treatment, etc.), a dimension they claim is a key part of their novelty.
- *Critically insufficient expert validation.* The validation process, particularly for a sensitive medical domain, is inadequate and falls well below standard scientific practice.
   - The utility of case reports is justified anecdotally via "informal interviews with 10 physicians," lacking any structured evidence or detailed findings.
   - The "distractor" modifications are a delicate process where errors could invalidate the answer. The complete list of modification types is not reported in the main paper, only two examples are reported. The authors state this stage was verified by three physicians for only an unspecified subset of questions, not the entire dataset. The subset size, selection criteria, provided instructions, verification results, and inter-annotator agreement are not provided.
   - Some details about instructions and annotations criteria are provided only for step 4 but, again, they appear incomplete and poorly designed. It seems that each report has been evaluated by one physician only. One quality dimension only (realism). As for the 1-5 Likert scale, the authors only provide definitions for 1 (unrealistic) and 5 (realistic), leaving intermediate values (2, 3, 4) to subjective interpretation.
- *Over-reliance on LLM-as-a-Judge.* The primary evaluation metric is an "LLM-as-a-Judge" (GPT-4o) for equivalence. Physician validation of this judge was performed on a very small subsample (45/235). While the 93% agreement is noted, the process, again, lacks detail.
- *Predictable RAG findings due to circular experimental design.* A primary conclusion is that RAG enhances performance. Assuming a successful retrieval and considering the nature of the applied modifications in QA generation, this finding is mostly an anticipated consequence of the experimental setup, where the benchmark's questions were derived directly from the retrieval corpus.
- *Questionable novelty of QA format.* The paper criticizes multiple-choice question benchmarks (e.g., MedQA, MedMCQA, PubMedQA) while advocating for its long-form questions and open-ended answers. However, the "open-ended" answers are exceptionally short (29 tokens on average), resembling simple verbalizations of a multiple-choice correct option. This raises doubts about whether this format truly evaluates the "open-ended reasoning" the authors claim is necessary. As other researchers have already created open-ended versions of benchmarks like MedQA by verbalizing correct answers with and without LLMs, the QA format, as implemented, does not appear to be a significant novel identity factor for this resource.
- *Insufficient depth in model and retriever analysis.* A deeper breakdown of error modes (e.g., where RAG fails despite correct retrieval, or which case types defeat open-source models) would be valuable.
- *Missing technical details.* No prompt or context engineering discussion, LLM decoding strategies information (crucial for results interpretation), and any statistical significance tests to validate the results.
- *Presentation quality.* The manuscript's quality is limited, featuring low-resolution, non-vectorial figures with default Draw.io colors, poorly organized tables, inconsistent notation (e.g., use of commas for thousands), and repeated acronym definitions.
- *Missing license.* The authors state the dataset and code will be publicly released but do not specify the license, which is a critical omission for a resource paper.
- *Domain drift and dataset maintenance.* In Appendix, the authors recognize that the benchmark's relevance may erode as some rare cases become standardized, yet there is limited technical prescription for maintaining dataset relevance or tracking drift over time.

**Questions:**

- How robust is the LLM-based evaluation metric to changes in prompting, underlying judge model, or model drift over time?
- Can the authors expand the analysis of error modes—for example, cases where retrieval finds the correct document but LLMs misinterpret, or vice versa?
- What is the observed impact of distractor additions on model/retriever confusion rates? Can the authors share specific ablations or examples where distractors led to errors, to clarify how realistic modifications challenge current systems?
- What steps could be taken to dynamically update or maintain the relevance and challenge of rare-case benchmarks as some cases become incorporated into guidelines?

---

> ### Author Response · Authors · 2025-11-21
>
> Thank you for your comments with great detail! We have a long response, so we will split it into multiple comments. References will be in the last comment.
>
>
> >Limited methodological novelty. The primary contribution is the use of case reports as a source, not the development of a novel benchmark methodology. The resource is constructed by filtering and sampling (with poor control) an existing PubMed Central corpus, applying simple
> semi-automatic LLM-based extraction, and performing (incomplete) manual verification.
>
> Thank you for giving us a chance to clarify our intention and contribution. While each step of dataset creation is standard, our novelty is introducing the task itself, which has not been studied before: answering open-ended off-guideline rare-case questions with a non-trivially created benchmark. Simple LLM-based extraction is not enough to create this benchmark. Multiple rounds of information extraction, verification, and modification were done for the creation of the dataset, as we described in section 3.1.
>
> Expert verification of all 235 question-answer pairs was done.
>
> >Superficial comparison to existing work. For a resource-centric paper, the comparison to the existing literature is insufficient. It lacks a detailed, quantitative comparison table positioning OGCAREBENCH against other benchmarks (especially those already using case reports) across key dimensions (e.g., scale, task, validation rigor). This makes it difficult to assess the true novelty or "delta" provided by this work.
>
> MedCaseReasoning [1] is the most similar work that uses case reports for benchmarking. This paper uses case reports to generate questions and asks for the diagnosis of the case presented along with the reason. This benchmark was created by filtering and extracting diagnosis and reason from the case report without multi-turn information extraction and modification. The greatest difference between this and our benchmark is that we focus on the significance, whether it is a diagnosis, a test, or a treatment that is novel. We focused on cases that are both rare and have at least one novelty. As case reports have an emphasis on novelty and rarity, or “significance”, our benchmark asks the essence of each case report. We also use modification methods to make our benchmark more realistic. If there are similar papers using case reports, could you kindly point us to them? This is by far the most similar paper according to our knowledge.
>
>
> >Questionable sampling and unbalanced dataset.
> Sampling. A "pure random sampling" of 1,100 reports (from which only 235 are finalized) is inadequate. Stratified sampling (e.g., by specialty, contribution type) would have been more rigorous. Balance. The dataset is highly unbalanced, with 70% of cases coming from only two specialties, a distribution that does not reflect the source corpus. The authors also fail to quantify the distribution of "contribution types" (diagnosis, treatment, etc.), a dimension they claim is a key part of their novelty.
>
> Our final dataset is indeed stratified. Table 1 shows the distribution of the source corpus and our benchmark, which are very similar.  The discipline distribution of the source corpus is also highly weighted on internal medicine and surgical studies. This is due to the nature of case reports and the two disciplines. Internal medicine is an overlapping field of various sub-specialties; each case is unique in surgical studies. Also, they are the major disciplines where doctors often consult case reports (line 137-139). This imbalance is expected and reflects the specialties where this benchmark is most applicable. Overall, the source corpus and final dataset have similar distribution (Table 1). Further details are in Section 3.2.
>
> The contribution type is distributed as follows:
> | Treatment | Test | Diagnosis |
> |-------------|-----|-----|
> | 183 | 31 | 21 |
>
> This is expected as many of the novelties lie in the scope of treatments, ranging from novel treatment or drug introduced to a new integrated multidisciplinary care strategy.

---

> > ### Author Response · Authors · 2025-11-21
> >
> > >Critically insufficient expert validation. The validation process, particularly for a sensitive medical domain, is inadequate and falls well below standard scientific practice.
> > The utility of case reports is justified anecdotally via "informal interviews with 10 physicians," lacking any structured evidence or detailed findings.
> >
> > The utility of case reports was investigated to scope the goal, not to validate a clinical procedure. It was an exploratory process to understand the domain and shape the task. Validation of the full dataset was done afterwards by physicians.
> >
> > >The "distractor" modifications are a delicate process where errors could invalidate the answer. The complete list of modification types is not reported in the main paper, only two examples are reported. The authors state this stage was verified by three physicians for only an unspecified subset of questions, not the entire dataset. The subset size, selection criteria, provided instructions, verification results, and inter-annotator agreement are not provided.
> >
> > We provided three examples of modification (demographics, terminology replacement, and comorbidities) and referred to the full prompt in the appendix for other modifications (line 253-255). Detailed examples in the instructions are: synonym, abbreviation, replace with common words, comorbidities, symptoms, tests, background information, past but resolved medical history, family history, etc.
> > The purpose of physician validation was to confirm the soundness of the modification to verify our methodology, not to verify every modification made. Verification of the full benchmark with the modification was done once we agreed on the method.
> >
> >
> >
> > >Some details about instructions and annotations criteria are provided only for step 4 but, again, they appear incomplete and poorly designed. It seems that each report has been evaluated by one physician only. One quality dimension only (realism). As for the 1-5 Likert scale, the authors only provide definitions for 1 (unrealistic) and 5 (realistic), leaving intermediate values (2, 3, 4) to subjective interpretation.
> >
> > Thank you for this observation. Quality dimensions are  “(1) the question and answer should be medically aligned, and (2) the question should require domain-specific medical expertise rather than general medical knowledge held by the public”, which is not restricted to realism.
> >
> > While intermediate values in the Likert scale may be subjective, the annotators commented on the question-answer pairs with a rating less than 4, explaining their reason for disapproval. We decided to use question-answer pairs with a rating of 4 or 5 after discussion with the annotators and going through their comments for lower-rated pairs.  Also, the annotators are clinically active, licensed physicians.
> >
> > >Over-reliance on LLM-as-a-Judge. The primary evaluation metric is an "LLM-as-a-Judge" (GPT-4o) for equivalence. Physician validation of this judge was performed on a very small subsample (45/235). While the 93% agreement is noted, the process, again, lacks detail.
> >
> > We provide the rationale and details for validation in number 2 of the general comment (LLM-as-a-judge validation and instructions for the annotators).

---

> > > ### Author Response · Authors · 2025-11-21
> > >
> > > >Predictable RAG findings due to circular experimental design. A primary conclusion is that RAG enhances performance. Assuming a successful retrieval and considering the nature of the applied modifications in QA generation, this finding is mostly an anticipated consequence of the experimental setup, where the benchmark's questions were derived directly from the retrieval corpus.
> > >
> > > The questions were significantly modified. The essence of the question remains the same, but the way of presentation is significantly different. This is an example of before and after modification.
> > >
> > >
> > > **Before modification**
> > >
> > > > A patient presented in May 2022 with depigmented patches on the hands and face. At that time, no treatment was initiated. By November 2022, the patient was referred to dermatology and diagnosed with active non-segmental, acrofacial vitiligo. On examination, depigmented patches were noted on the hands and face, and treatment with tacrolimus 0.1% ointment, applied twice daily, was initiated. Between November 2022 and March 2023, the patient developed new and expanding depigmented macules and patches, prompting the addition of oral mini-pulse prednisone (10 mg for two consecutive days per week). The patient also intermittently used ginkgo biloba (120 mg). After 4 months of treatment, at the March 2023 follow-up, partial improvement was observed in larger patches, but progress plateaued by the second month. New depigmented macules had appeared on the jawline and scalp. The patient expressed dissatisfaction with the treatment regimen due to the greasy texture of the ointment, the inconvenience of multiple oral treatments, and side effects such as gastrointestinal upset and elevated blood sugar levels. Given the patient's persistent disease activity and dissatisfaction with the current regimen, what would be the most appropriate next step in management?
> > >
> > > **After modification**
> > >
> > > > A 42-year-old patient initially noticed hypopigmented lesions affecting bilateral dorsal hands and perioral regions in April 2022, though no intervention was pursued at that time. The patient has a history of well-controlled type 2 diabetes mellitus on metformin 500mg twice daily and seasonal allergies managed with loratadine as needed. Following dermatological consultation in October 2022, the diagnosis of progressive generalized vitiligo with acrofacial distribution was confirmed. Wood's lamp examination revealed characteristic enhancement of the achromic areas. Initial therapy consisted of tacrolimus 0.1% topical immunomodulator applied twice daily to affected areas. During the treatment period from October 2022 through February 2023, the patient experienced progression with emerging and enlarging achromic lesions, leading to supplementation with systemic corticosteroid pulse therapy (prednisolone 10mg on two sequential days weekly). The patient's mother has a history of autoimmune thyroiditis, though the patient's thyroid function tests remain normal. The patient self-administered ginkgo biloba extract (120mg) sporadically based on internet research. Recent laboratory work showed normal CBC, liver enzymes, and vitamin D levels. Following 16 weeks of combination therapy, the February 2023 assessment revealed modest response in established lesions, though therapeutic benefit reached a plateau after 8 weeks. Fresh hypopigmented areas emerged along the mandibular region and vertex scalp, with some follicular repigmentation noted in older lesions. The patient reported treatment-related challenges including the unpleasant consistency of topical preparations, complexity of the medication schedule, digestive disturbances, and hyperglycemia requiring adjustment of diabetes medications. The patient works as a teacher and finds the visible lesions psychologically distressing. Considering ongoing disease progression and patient's treatment intolerance, what therapeutic modification would be most suitable?

---

> ### Author Response · Authors · 2025-11-21
>
> >Questionable novelty of QA format. The paper criticizes multiple-choice question benchmarks (e.g., MedQA, MedMCQA, PubMedQA) while advocating for its long-form questions and open-ended answers. However, the "open-ended" answers are exceptionally short (29 tokens on average), resembling simple verbalizations of a multiple-choice correct option. This raises doubts about whether this format truly evaluates the "open-ended reasoning" the authors claim is necessary. As other researchers have already created open-ended versions of benchmarks like MedQA by verbalizing correct answers with and without LLMs, the QA format, as implemented, does not appear to be a significant novel identity factor for this resource.
>
> Using the same tokenizer used to tokenize our examples, MedQA-USMLE [5] test set has an average answer choice length of 6.69 tokens and an average question length of 33.5 tokens. This dataset has longer answer choices than other renowned datasets, such as MedMCQA [6], which has answer choices of a couple words and PubMedQA’s [2] answer choices of yes/no/maybe.  Yet, this is significantly shorter than our benchmark. The average length of questions is also shorter compared to our questions (avg. 403.3 tokens), which are based on abnormal cases rather than standard cases.
> Additionally, our questions require significantly more reasoning than open-ended versions of these benchmarks, which rephrase a single correct option. For example, if the answer was option A, the full text of option A would become the answer to the open-ended version. Open-ended benchmarks are often consumer-based or test simple medical knowledge rather than a case-by-case scenario with abnormality. We are not claiming novelty in the QA format, but we rather want to emphasize the amount of reasoning and information required to derive an answer to our question.
>
>
> >Insufficient depth in model and retriever analysis. A deeper breakdown of error modes (e.g., where RAG fails despite correct retrieval, or which case types defeat open-source models) would be valuable.
>
> >Can the authors expand the analysis of error modes—for example, cases where retrieval finds the correct document but LLMs misinterpret, or vice versa?
>
> Thank you for raising this. For failure analysis, we created a contingency table of retrieval and RAG. Please refer to number 3 of the general comment (Contingency table of retrieval and RAG) for the table and analysis. We aim to further look into retrieval success yet having incorrect RAG responses, focusing on criteria such as specialty or the significance type of case report (treatment, diagnosis, test).
>
>
> >Missing technical details. No prompt or context engineering discussion, LLM decoding strategies information (crucial for results interpretation), and any statistical significance tests to validate the results.
>
> Thank you for raising this. The prompts are in Appendix C. For any applicable model, temperature was set to 0.7, top_p to 0.95, and max_tokens to 2000.
>
> >Missing license. The authors state the dataset and code will be publicly released but do not specify the license, which is a critical omission for a resource paper.
>
> Thank you for bringing this up. We added a license and updated supplementary material.
>
> >Domain drift and dataset maintenance. In Appendix, the authors recognize that the benchmark's relevance may erode as some rare cases become standardized, yet there is limited technical prescription for maintaining dataset relevance or tracking drift over time.
>
> >What steps could be taken to dynamically update or maintain the relevance and challenge of rare-case benchmarks as some cases become incorporated into guidelines?
>
>
> Thank you for raising this reasonable concern. Upon acceptance, we plan to publicize the full code and prompts used for the generation of our benchmark so that it can be updated as knowledge drifts over time. The pipeline is not hard to reproduce, and PMC allows mass public download of their papers via FTP. However, outdating of the dataset is not unique to our benchmark, but rather an issue for most benchmarks. This is also acknowledged in the limitations section.

---

> > ### Author Response · Authors · 2025-11-21
> >
> > >How robust is the LLM-based evaluation metric to changes in prompting, underlying judge model, or model drift over time?
> >
> > We provide the prompt for LLM-as-a-judge evaluation. While we have a uniform prompt and model, we expect evaluation quality to be improved as LLMs advance. Using SOTA LLMs as a judge will ensure reliable evaluation quality.
> >
> > >What is the observed impact of distractor additions on model/retriever confusion rates? Can the authors share specific ablations or examples where distractors led to errors, to clarify how realistic modifications challenge current systems?
> >
> > Without distractors, retrieval performance is high, approaching nearly 1, as the questions are directly from the case reports without modification. Distractors were validated during the annotation of the full dataset, and 53 questions were dropped.
> >
> > **References**
> >
> > [1] Wu, Kevin, et al. "MedCaseReasoning: Evaluating and learning diagnostic reasoning from clinical case reports." arXiv preprint arXiv:2505.11733 (2025).
> >
> > [2]Jin, Qiao, et al. "Pubmedqa: A dataset for biomedical research question answering." Proceedings of the 2019 conference on empirical methods in natural language processing and the 9th international joint conference on natural language processing (EMNLP-IJCNLP). 2019.
> >
> > [3]​​Gupta, Deepak, et al. “A Dataset of Medical Questions Paired with Automatically Generated Answers and Evidence-supported References.” Scientific Data, vol. 12, 2025, article no. 1035. Nature Publishing Group, https://doi.org/10.1038/s41597-025-05233-z
> >
> > [4] Zheng, Lianmin, et al. "Judging llm-as-a-judge with mt-bench and chatbot arena." Advances in neural information processing systems 36 (2023): 46595-46623.
> >
> > [5] Jin, Di, et al. "What disease does this patient have? a large-scale open domain question answering dataset from medical exams." Applied Sciences 11.14 (2021): 6421.
> >
> > [6] Pal, Ankit, Logesh Kumar Umapathi, and Malaikannan Sankarasubbu. "Medmcqa: A large-scale multi-subject multi-choice dataset for medical domain question answering." Conference on health, inference, and learning. PMLR, 2022.

---

> ### Author Response · Authors · 2025-11-26
>
> **Dear Reviewer wxnk, Thank you again for your review of our work. We understand you might be busy, but we hope you will be able to review our rebuttal. We are happy to clarify or elaborate on any points as needed. If our response addresses your concerns and positively influences your view of the work, we would be grateful if you reflect that in an updated score.**

---

### Official Review · Reviewer_hJvx · 2025-11-04

**Soundness:** 3
**Presentation:** 3
**Contribution:** 2
**Rating:** 6
**Confidence:** 4

**Summary:**

This paper introduces OGCaReBench, i.e. a benchmark to evaluate LLM and RAG performance on questions derived from rare clinical case studies. To this end, a dataset is constructed semi-automatically by mining publicly available case reports, converting them into QA format and altering non-relevant parts of the report. A suite of LLMs and retrievers is evaluated, suggesting that LLMs alone struggle with the benchmark, as evaluated by a (validated) judge LLM, but retrieval improves performance - with access to original documents which the QA pair is derived from, accuracy is very high, suggesting that the problem is mostly that of retrieval.

**Strengths:**

The paper is well written, i think overall the research is largely well executed. I appreciate the validation of LLM results with human annotators.

**Weaknesses:**

I find no big weaknesses with the execution of the research, only a few remarks:

- it would be great to have more details regarding the protocols used for human validation (both the validity of the QA pairs as well as the LLM as a judge).

- It seems like the problem is mostly with retrieval - given that the larger LLMs have very big context windows, I'm not sure why the RAG experiments reported in table 7 stop at 5 documents. It would be good to see results if the context window is maxxed out for each LLM.

- Given the rather small dataset size, it would be good to have statistical significance reporting, in form of statistical significance tests as well as confidence intervals, to contextualise the amount of statistical uncertainty

That being said, I do question the motivation and therefore the overall contribution of the paper a bit.

The use case appears to be very applied, therefore it should be of interest for domain experts who could use such technology (i.e. physicians). But then, QA is only a proxy of the actual task at hand, that is helping to find the appropriate course of action for a patient, which shouldn't be evaluated in QA format but rather as the actual task, where physicians are (or aren't) supported by technology (such as LLMs) to inform their action. Probably, the findings would be of more interest for audiences of medical (informatics) journals, who could also more rigorously judge the significance and validity of the research, rather than an AI conference.

Looking at the QA formulation, the findings are somewhat plain - LLMs struggle to answer questions about rare cases and using RAG (and retrieving very similar cases) improves performance. I don't find this finding particularly exciting. In order to tease interest for more AI/CS relevant audiences, it would be good to see analyses investigating the root causes, failure modes and potential avenues for or actual demonstrated methods of improvement.

**Questions:**

Please address my minor remarks stated in the weaknesses.

Also, did you tamper with the submission template? The margins are smaller compared to the official template.

**Details Of Ethics Concerns:**

no report how annotators were compensated

---

> ### Author Response · Authors · 2025-11-21
>
> Thank you for highlighting our strengths and thoughtful questions!
>
> >it would be great to have more details regarding the protocols used for human validation (both the validity of the QA pairs as well as the LLM as a judge).
>
> Thank you for this comment. Human validation is indeed an important procedure. We have our details and prompt about using validating LLM-as-a-judge on number 2 of the general comment (LLM-as-a-judge validation and instructions for the annotators).
>
> The instructions for validating QA pairs are in Figure 11. We focus on the clinical validity and realism of the question and answer, and whether it requires expert-level knowledge. The annotators are clinically active, licensed physicians in internal medicine and oncology.
>
> >It seems like the problem is mostly with retrieval - given that the larger LLMs have very big context windows, I'm not sure why the RAG experiments reported in table 7 stop at 5 documents. It would be good to see results if the context window is maxxed out for each LLM.
>
> Retrieval is important to our benchmark, so testing greater contexts is a valid suggestion. However, major increases according to context length occur between non-RAG to RAG and using 1 context document to 3 documents. Using 5 context documents usually increases accuracy by 2-3% and in some cases, it has lower performance than using 3 contexts (MedGemma, BMReteriever with GPT5). Given this, we did not further increase the context as it could introduce noise rather than provide noticeable improvements.
>
> > Given the rather small dataset size, it would be good to have statistical significance reporting, in form of statistical significance tests as well as confidence intervals, to contextualise the amount of statistical uncertainty
>
> Thanks for the comment. We did a statistical analysis for this, which is number 1 in the general comment (Statistical analysis for dataset size). According to its result, our benchmark of 235 QA pairs provides a stable performance estimate.
>
> >The use case appears to be very applied, therefore it should be of interest for domain experts who could use such technology (i.e. physicians). But then, QA is only a proxy of the actual task at hand, that is helping to find the appropriate course of action for a patient, which shouldn't be evaluated in QA format but rather as the actual task, where physicians are (or aren't) supported by technology (such as LLMs) to inform their action. Probably, the findings would be of more interest for audiences of medical (informatics) journals, who could also more rigorously judge the significance and validity of the research, rather than an AI conference.
>
> Our goal is to introduce a controlled and reproducible task to evaluate LLM performance in the medical domain beyond general medical knowledge and the official guidelines that can be memorized, rather than to evaluate clinical procedures directly. We want to introduce a new challenge in the NLP community in terms of domain-specific tasks. As direct evaluation of models in real-world clinical procedures is infeasible and cannot be easily reproduced, we hope our benchmark works as a simulator of treating rare cases where high-level medical reasoning is necessary.
>
> >Looking at the QA formulation, the findings are somewhat plain - LLMs struggle to answer questions about rare cases and using RAG (and retrieving very similar cases) improves performance. I don't find this finding particularly exciting. In order to tease interest for more AI/CS relevant audiences, it would be good to see analyses investigating the root causes, failure modes and potential avenues for or actual demonstrated methods of improvement.
>
> The empirical results show the gap that was not previously identified. We identify the shortcomings of SOTA LLMs in previously unseen medical questions requiring deep reasoning unless supported by retrieval, even though their performance against other medical benchmarks is reported to be good. Also, renowned Medical benchmark datasets have been previously accepted in AI venues: PubMedQA - EMNLP 2019 [1], MedXpertQA - ICML 2025 [2]
>
> For failure analysis, we created a contingency table of retrieval and RAG. Please refer to number 3 of the general comment (Contingency table of retrieval and RAG) for the table and analysis. We aim to further look into retrieval success, yet having incorrect RAG responses, focusing on criteria such as specialty or the significance type of case report (treatment, diagnosis, test)
>
> We checked the submission template. No change was made to the style.
>
> **References**
>
> [1] Jin, Qiao, et al. "Pubmedqa: A dataset for biomedical research question answering." Proceedings of the 2019 conference on empirical methods in natural language processing and the 9th international joint conference on natural language processing (EMNLP-IJCNLP). 2019.
>
> [2] Zuo, Yuxin, et al. "Medxpertqa: Benchmarking expert-level medical reasoning and understanding." arXiv preprint arXiv:2501.18362 (2025).

---

> ### Author Response · Authors · 2025-11-26
>
> **Dear Reviewer hJvx, Thank you again for your review of our work. We understand you might be busy, but we hope you will be able to review our rebuttal. We are happy to clarify or elaborate on any points as needed. If our response addresses your concerns and positively influences your view of the work, we would be grateful if you reflect that in an updated score.**

---

### Author Response · Authors · 2025-11-21
**General comments**

**1. Statistical analysis for dataset size**

We used bootstrap analysis, which is a resampling method of repeatedly drawing samples with replacement to approximate the sampling distribution, to confirm that our benchmark shows stable results. 1000 bootstrap sampling of both the baseline and BGE retriever with 3 reports as context confirms that our benchmark has a stable performance estimate. The bootstrap mean is nearly identical to the reported accuracy. This shows that our clinically rich benchmark of 235 questions is sufficient for model comparison.

**Baseline**
| Model | Reported Accuracy | Bootstrap Mean | 95% CI |
|-------|------|------|--------|
| GPT-5 | 0.447 | 0.447 | 0.383–0.515 |
| GPT-o3-mini | 0.515 | 0.512 | 0.447–0.574 |
| MedGemma-27b-text-it | 0.362 | 0.361 | 0.302–0.421 |
| Llama3-Med42-70B | 0.421 | 0.421 | 0.357–0.485 |

**Retriever: BGE, Context Length: 3 reports**
| Model | Reported Accuracy | Bootstrap Mean | 95% CI |
|-------|------|------|--------|
| GPT-5 | 0.728 | 0.728 | 0.672–0.783 |
| GPT-o3-mini | 0.719 | 0.720 | 0.664–0.779 |
| MedGemma-27b-text-it | 0.630 | 0.629 | 0.570–0.694 |
| Llama 3-Med42-70B  | 0.562 | 0.562 | 0.498–0.626 |


**2. LLM-as-a-judge validation and instructions for the annotators**

Following is the instruction for the human annotators for the LLM-as-a-judge quality validation.

This was the instruction to the model:

>- Consider them equivalent if they refer to the same kind of medical action, even if wording differs. If the main medical procedure is similar and other details somewhat aligns, two texts are equivalent.
>- Mark as mismatch if the response suggests a different type or intent of medical procedure. If two texts include similar medical procedure but their importance differs greatly, or their main medical procedure differs, mark as mismatch. If one suggests a broad method and the other specifically mention one of the methods involved in the broad method, mark as mismatch.
>- Some texts have reasons or rationale explaining their main content. Do not use this part to determine equivalence or mismatch.
>
>Please comment on the annotation column if this content is labeled correctly according to the description above. It will be a binary classification of yes/no.

Using the Wilson confidence interval, the 95% confidence interval is 82.1%-97.7%. This surpasses the agreement levels in MT-Bench [1], which has human-human agreement of approximately 81% and LLM-human agreement of above 80%.


**3. Contingency table of retrieval and RAG**

We conducted further analysis by jointly examining retrieval success and RAG answer correctness. This is the result for BGE retriever and 3 context documents, and we are planning to add results and analysis for the full set.
| Model | Total | RAG Accuracy | Ret. Succ. + RAG Correct | Ret. Fail + RAG Correct | Ret. Succ + RAG Incorrect | Ret. Fail + RAG Incorrect |
|-------|-------|---------|------------|------------|------------|------------|
| GPT-5 | 235 | 72.8 | 144 | 27 | 32 | 32 |
| GPT-o3-mini | 235 | 71.9 | 137 | 32 | 39 | 27 |
| Llama 3.3 70B Instruct | 235 | 66.0 | 131 | 24 | 45 | 35 |
| Claude 4 Sonnet | 235 | 67.7 | 135 | 24 | 41 | 35 |
| Thinking Claude 4 Sonnet | 235 | 73.6 | 144 | 29 | 32 | 30 |
| MedGemma-27b-text-it | 235 | 63.0 | 122 | 26 | 54 | 33 |
| Llama 3-Med42-70B | 235 | 56.2 | 111 | 21 | 65 | 38 |
| OpenBioLLM-Llama 3-70B | 235 | 48.1 | 94 | 19 | 82 | 40 |

This is a part of failure analysis using retrieval success and RAG success. In general, <Retrieval Success + RAG Incorrect> is greater than <Retrieval Fail + RAG Correct> or <Retrieval Fail + RAG Incorrect>, suggesting that medical reasoning is a challenging task even when a correct document is provided. Models with good performance (GPT-5, Thinking Claude 4 Sonnet) have low <Retrieval Success + RAG Incorrect>, implying that pointing to the correct document often results in an accurate answer if the model’s performance is good. Their high <Retrieval Fail + RAG Correct> also emphasizes the necessity of reasoning ability in answering challenging medical questions, which aligns with our findings in the paper.

**References**

[1] Zheng, Lianmin, et al. "Judging llm-as-a-judge with mt-bench and chatbot arena." Advances in neural information processing systems 36 (2023): 46595-46623.

---

### Meta-Review · Area_Chair_GGag · 2026-01-11

**Summary:**

This paper presents a medical QA benchmark for evaluating existing LLMs on "off-guideline" clinical scenarios—cases that are rare, novel, or complex enough that they are not covered by standard medical pathways.The authors filtered ~53k PubMed Central case reports, used an LLM-based pipeline to extract 1,100 QA pairs, and applied "distractor" modifications (altering demographics and terminology) to prevent simple keyword matching. A final set of 235 QA pairs was validated by physicians for clinical realism and difficulty. The paper reports that even top-tier models like GPT-o3-mini only achieve ~51% accuracy in a zero-shot setting, while Retrieval-Augmented Generation (RAG) using the source corpus improves performance to ~75%. The authors argue that this benchmark is a necessary simulator for real-world clinical consultation where doctors must look beyond memorized guidelines.
The reviewers raised several substantive concerns. These include questions about the paper’s novelty and its suitability for the ICLR audience, the limited dataset size (235 QA pairs), potential circularity and contamination risks in the experimental design, the appropriateness of case reports as a gold standard for evaluation, and the heavy reliance on LLM-as-a-judge, which may undermine the benchmark’s reliability. The initial review scores were 2, 2, 4, and 6, and these scores remained unchanged during the discussion period. Several reviewers did not engage in the discussion, while one engaged reviewer maintained a strongly negative assessment (score 2).

After carefully considering the reviews and the rebuttal, the AC agrees that the paper explores an interesting and relevant direction—namely, the evaluation of LLMs on rare or atypical clinical cases. However, significant concerns remain. In particular, even with the presented bootstrap analysis over 1,000 samples, it is difficult to justify that a benchmark of 235 QA pairs is sufficient to meaningfully assess rare clinical phenomena. Moreover, the assumption that less-cited publications are more likely to contain rare cases is not convincingly justified and requires stronger empirical or clinical grounding. While several other concerns were partially addressed in the rebuttal, these core issues remain unresolved.

That said, the AC believes that the topic itself—benchmarks targeting off-guideline or rare clinical reasoning—is well aligned with the ICLR dataset and benchmark track. Nevertheless, given the lack of reviewer engagement during discussion and the persistence of key concerns, the AC anticipates that the reviewer scores would largely remain unchanged even with fuller participation. As such, the paper remains below the acceptance threshold. Overall, the AC believes the work would benefit from more rigorous analysis and stronger evidence demonstrating that the proposed benchmark meaningfully aligns with real-world clinical reasoning and evidence, beyond simply extracting cases from less-cited PubMed articles.

**Reviewer Concerns:**

The major three concerns which seem to be well addressed are: 1. The RAG task was trivial because questions were derived from the same papers. The authors provided a "Before vs. After" modification example, proving that significant changes to patient demographics, terminology, and comorbidities forced the models to reason rather than just keyword-match. 2. How to make sure the LLM is not overly replied upon: To counter concerns about GPT-4o being a biased judge, the authors reported a 93.3% agreement rate with licensed physicians. They used a Wilson confidence interval to prove that this judge is mathematically as reliable as, or more reliable than, typical human-human agreement in NLP. 3. How to make sure the dataset is balanced: The authors demonstrate that the imbalance is expected and reflects the specialties where this benchmark is most applicable. For example, the focus on Internal Medicine and Surgery (70%) was not a sampling error but a reflection of the domain, as these are the specialties that most frequently consult case reports in practice. Other issues such as whether the case report can be gold standard and the failure analysis were addressed.

Other concerns not fully addressed: Reviewers expressed concern that the dataset size of 235 QA pairs is too limited to support strong conclusions. Although the authors provided a bootstrap analysis over 1,000 samples suggesting that accuracy estimates are stable and that the benchmark can distinguish between models, this evidence is not fully convincing. Because the analysis is conducted on QA pairs derived from low-citation PubMed articles, it remains unclear whether the observed stability meaningfully reflects coverage of genuinely rare or clinically challenging cases. In particular, the assumption that less-cited publications are more likely to contain rare cases is not sufficiently justified and would require stronger empirical or clinical validation. Additionally, the authors argue that establishing a human performance baseline is “hard and time-consuming.” However, in the context of ICLR—where benchmarks are expected to be carefully calibrated and interpretable—the absence of a human baseline significantly limits the benchmark’s usefulness. Without a human–AI comparison, it is difficult to assess whether the reported LLM performance represents a meaningful challenge or an unrealistically low bar.

**Reviewer Scores:**

The initial review scores were 2, 2, 4, and 6. Given the limited reviewer engagement during the discussion period—particularly the lack of resolution on shared concerns such as the small dataset size of 235 QA pairs (as discussed above)—these scores are unlikely to have changed even if they all had been able to participate fully in the discussion. One engaged reviewer maintained a strongly negative assessment (score 2) and did not provide further feedback after the authors offered additional justification. Therefore, the final score is largely likely to be still below the acceptance.

---

### Decision · Program_Chairs · 2026-01-26

Reject